# Thalamocortical feedback selectively controls pyramidal neuron excitability

Federico Brandalise[1,3,4], Ronan Chéreau [1,4], I-Wen Chen [1], David van Oorschot [1], Claudia Morin Raig [1], Tanika Bawa [1], Nandkishor Mule [1], Stéphane Pagès[1,2], Foivos Markopoulos[1] & Anthony Holtmaat [1] ✉

The apical dendrites of layer (L) 2/3 pyramidal neurons in the mouse somatosensory cortex integrate synaptic input from long-range projections. Among those, inputs from the higher-order thalamic posteromedial nucleus may facilitate sensory-evoked cortical activity, but it remains elusive how this role emerges. Here we show using ex vivo dendritic recordings that these projections provide dense synaptic input to broad tufted neurons residing predominantly in L2 and cooperate with other inputs to produce NMDA spikes. They have the unique capacity to block two-pore domain potassium leak channels via group 1 metabotropic glutamate receptor (mGluRI) signaling, which increases excitability. Slender tufted L2/3 neurons and other long-range projections fail to invoke these mechanisms. In vivo imaging of calcium signals confirms the presence of mGluRI-dependent modulation of feedback-mediated spiking in L2. Our results imply that higher-order thalamocortical projections regulate neuronal excitability in a cell type and input-selective manner through fast NMDAR and mGluRI-dependent mechanisms.

The neocortex consists of an intricate network of feedforward and feedback connections, but their topology and functional interactions remain enigmatic. Cortical pyramidal neurons receive distinct synaptic input, depending on their subtype and laminar location[1]. A striking example of spatially segregated inputs to pyramidal neurons is formed by first-order and higher-order parallel thalamocortical projections. In the mouse primary somatosensory cortex (S1), these projections are broadly yet distinctly distributed over all cortical layers[2]. First-order thalamocortical projections from the ventroposterior medial (VPM) thalamus target layer (L) 4 neurons and basal dendrites of thick-tufted L5b and L3 pyramidal neurons, whereas higher-order thalamocortical projections from the posteromedial nucleus (POm) mainly project to L5 and L1, targeting basal dendrites and apical tufts of L5a as well as the apical dendrites of L2/3 pyramidal neurons[3–6]. Within the L2/3 population, these inputs might be biased to L2 neurons[6,7], but it remains unclear which factors determine this connectivity.

L2/3 pyramidal neurons comprise a morphologically heterogeneous population, with neurons in L2 often bearing extensive apical dendritic tufts, known as broad tufted (BT) neurons, and those in L3 with small tufts, known as slender tufted (ST) neurons[8,9]. However, in mice, the laminar position of pyramidal neurons does not strictly correlate with tuft complexity[10]. The different arrangements of L2/3 pyramidal neuron dendrites are likely to translate into differences in their connectivity, which typically correlates with the amount of axodendritic overlap formulated as Peters' rule[11,12]. Therefore, L2 BT neurons may receive more inputs from long-range axons in L1, whereas L3 ST neurons with disproportionally more basal dendrites may receive biased input from local axons and those terminating in L4 and L3[13–15]. Accordingly, some L2 neurons have been shown to receive relatively strong input from POm thalamocortical projections[7], although this may not be a general principle for all L2 neurons[6]. Peters' rule does not apply to all cortical networks. For example, intracortical connectivity

[1]Department of Basic Neurosciences and the Geneva University Neurocenter, Centre Médical Universitaire (CMU), University of Geneva, Geneva, Switzerland. [2]WYSS Center, Campus Biotech, Geneva, Switzerland. [3]Present address: Division of Neuroscience and Clinical Pharmacology, Department of Biomedical Sciences, University of Cagliari, Cagliari, Italy. [4]These authors contributed equally: Federico Brandalise, Ronan Chéreau. ✉e-mail: anthony.holtmaat@unige.ch

of L2 pyramidal neurons is sometimes higher than predicted by axo-dendritic overlap, whereas the input from POm to L5b pyramidal neuron apical tufts in L1 is lower than expected[4,16]. Therefore, it remains unclear if axo-dendritic overlap is a good indicator for the input that L2/3 pyramidal neurons receive from POm afferents.

The connectivity patterns of POm projections suggest that they have distinct roles in the cortical circuitry. This is supported by the notion that synaptic responses evoked by higher-order thalamocortical projections such as from POm, have signatures that are different from synaptic responses elicited by first-order thalamocortical or corticocortical projections[17,18].

Glutamatergic pathways can be categorized into two groups, termed "drivers" and "modulators." Driver pathways, such as the pathway from VPM to S1, are linked to information-bearing pathways, whereas modulator pathways, such as the pathway from POm to S1, modify these primary information streams[19,20]. One distinction pertains to the presence of a metabotropic glutamate receptor (mGluR) component[21,22], but it remains enigmatic how this affects synaptic integration in L2/3 pyramidal neurons. It has been proposed that POm facilitates sensory-evoked responses of pyramidal neurons subpopulations by eliciting long-lasting depolarizations[23–28], but their underlying mechanisms also remain largely unknown.

Here, by combining electrophysiological dendritic patch-clamp recordings and optogenetics we show that L2/3 neurons with morphologically different dendritic trees receive biased inputs from long-range and local corticocortical circuits. POm thalamocortical synaptic inputs are dense on L2/3 BT neurons, whereas VPM thalamocortical synapses are biased to L2/3 ST neurons. BT neurons produce N-methyl-D-aspartate (NMDA) spikes when POm thalamocortical afferents are stimulated together with other afferents. In addition, we found that POm thalamocortical inputs are unique in their ability to elicit delayed sustained dendritic potentials (DSDPs) in BT neurons. This effect is mediated by the activation of group 1 mGluRs (mGluRI), which through an interaction with two-pore domain potassium (K2P) leak channels increase the local membrane input resistance. Using 2-photon laser scanning microscopy of calcium signals in vivo, we confirm that movement-related activity in these neurons, which is associated with recruitment of feedback circuits, is modulated through mGluRI-mediated mechanisms. We propose that higher-order thalamocortical projections regulate cortical sensory processing by gating the excitability of subpopulations of pyramidal neurons through fast and reversible NMDA receptor (NMDAR) and mGluRI-dependent mechanisms.

## Results

### Distinct long-range inputs to morphological subtypes of L2/3 pyramidal neurons

To compare the relative net input of various long-range and local afferents into putatively different types of L2/3 pyramidal neurons, we expressed genetically encoded opsins (ChR2 or ChrimsonR) in putative synaptic afferents using adeno-associated viral (AAV) vectors and recorded from L2/3 pyramidal neuron dendrites in brain slices. During the recordings, cells were filled with biocytin which allowed us to reconstruct the morphology of 27 cells for further analysis. To determine the spatial organization of the L2/3 pyramidal neuron dendrites in L1 (Fig. 1a and Supplementary Fig. 1) we measured the dendritic density and the span of the tree within the most superficial 200 μm of the somatosensory cortex. Using k-means clustering, the neurons were segregated into two groups (Fig. 1b), one with reduced and narrow dendritic trees, and one with dense and laterally spreading dendritic trees in L1. The first group typically exhibited a main apical branch extending perpendicular to the pia (Supplementary Fig. 1a), bearing similarities to the formerly reported ST neurons. The second group had dendrites that often originated from two main branches extending in an oblique way towards the pia, similar to the so-called BT neurons[8,14,29] (Supplementary Fig. 1a). In accordance, both the total

length and the number of branches of apical dendrites was larger in BT than in ST neurons whereas the number of branches of basal dendrites was larger in ST than in BT neurons (Supplementary Fig. 1b, c). This resulted in a greatly different apical-to-basal ratio of both branch length and number between the two types (Fig. 1c, d), which subsequently allowed us to classify neurons as BT or ST without extensive quantitative reconstructions. In addition, although individual neurons could not be classified as BT or ST neurons based on their laminar position (Supplementary Fig. 1a), the BT neurons were on average located more superficially as compared to ST neurons (Fig. 1e), in accordance with previous observations[14]. Pia-aligned dendritic density heatmaps of the two groups indicate that at the population level BT neurons have an overall higher density of dendrites in the L1-L2 region of cortex, whereas ST neurons have slightly more dendritic coverage in L3 (Fig. 1f, see also ref. 30). Passive electrophysiological parameters of BT and ST neurons were comparable, apart from the slow time constant and capacitance, which was likely a consequence of the morphological differences (Supplementary Fig. 1d, e).

To estimate the potential synaptic connectivity between long-range thalamocortical or corticocortical afferents and the two types of pyramidal neurons, we compared the laminar distribution of the opsin-labeled axons relative to the reconstructed dendritic trees (Fig. 1g–i). Thalamocortical afferents from POm and corticocortical afferents from the primary motor cortex (M1) and secondary somatosensory cortex (S2) overlapped more with BT dendrites, whereas thalamocortical afferents from VPM overlapped somewhat more with ST dendrites (Fig. 1j).

According to Peters' rule[11,12], the patterns of overlap as depicted in Fig. 1 predict that BT neurons receive relatively more synaptic inputs from POm, M1 and S2 than ST neurons do, and vice versa, ST neurons should receive relatively more input from VPM in comparison to BT cells. We tested this hypothesis by recording postsynaptic potentials (PSPs) at the main apical dendrite of BT and ST neurons while photostimulating the various inputs using 5-ms light pulses (for labeling and recording strategies, see Methods; Fig. 2a). Dendritic recordings were used since these are well suited for observing distal dendritic depolarization which readily attenuates toward the soma[31–34]. To account for the variability in the opsin expression levels and patterns over different preparations, we aimed at including both types in each slice to measure the relative synaptic input strength (Supplementary Fig. 2a). Together, this allowed comparisons of input strength from a particular afferent between nearby BT and ST neurons that were surrounded by a similar density of opsin-expressing axons (Supplementary Fig. 2b). The evoked PSP amplitudes increased monotonically with the amount of ChR2-GFP or ChrimsonR-tdTomato fluorescence (Supplementary Fig. 2c). Photo-stimulation of POm afferents evoked PSPs with higher amplitudes in BT neurons as compared to ST neurons (Fig. 2b). Conversely, stimulation of VPM afferents evoked larger PSPs in ST neurons as compared to BT neurons. Stimulation of M1 and S2 afferents did not result in statistically different PSP amplitudes between the two types. We also tested the input strength from intracortical S1 inputs (S1$_{intracortical}$) and found no significant differences between the two types (Fig. 2b). The PSP rise times were not different between the two types under any of the stimulation conditions (Supplementary Fig. 2d), indicating that the different PSP amplitudes between BT and ST neurons were not due to variations in the distance between the synaptic inputs and the recording sites. To verify that the observed POm and VPM-evoked PSPs included monosynaptic inputs, we bath-applied TTX and 4-AP in a subset of the recordings[4] (Supplementary Fig. 3a). The application did not abolish POm-evoked PSPs in BT and ST neurons, and VPM-evoked PSPs remained present in ST neurons (Supplementary Fig. 3b). However, the VPM-evoked PSPs in BT neurons were reduced to baseline noise-levels. This data indicates that the responses in ST neurons included monosynaptic PSPs from both POm and VPM, but that BT neurons only receive detectable

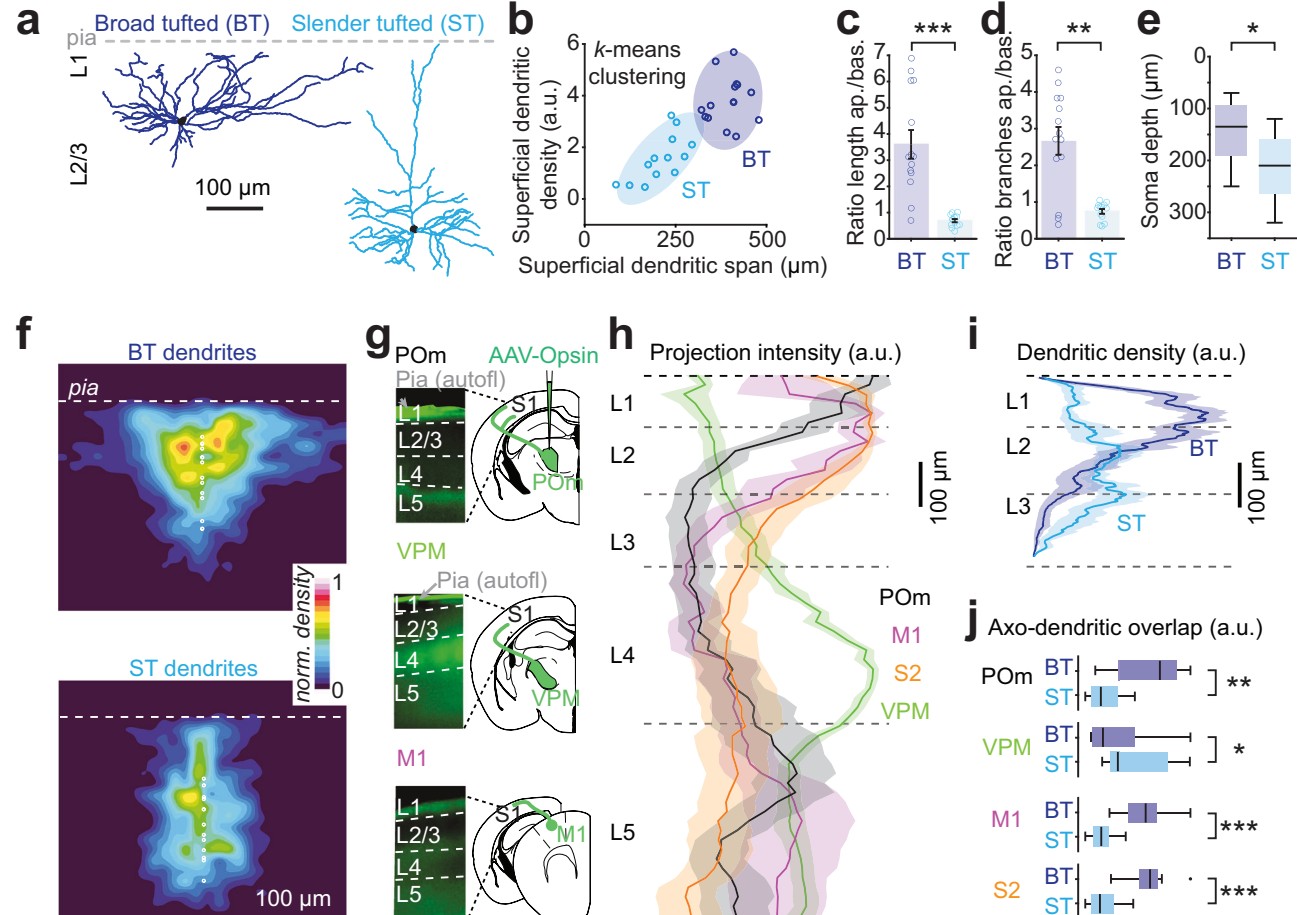

**Fig. 1 | BT and ST neurons in S1 form distinct groups based on morphological features. a** Examples of morphological reconstruction from biocytin-filled L2/3 pyramidal neurons in S1 indicating at least two distinct groups of L2/3 pyramidal neurons as previously described[8,9]: BT neurons exhibiting a large and dense apical arborization and ST neurons displaying a reduced apical arborization. **b** 27 neurons are segregated into BT and ST using a *k*-means clustering method (with *k* = 2) by comparing the dendritic span and density within the first 200 μm from the pia. Ellipsoid areas represent the 95% confidence interval of both clusters. **c** Ratio of the apical over basal dendritic length for BT and ST neurons (*n* = 14 BT neurons and 13 ST neurons, *P* = $5.1 \times 10^{-5}$, two-sided Wilcoxon rank-sum test). **d** Ratio of the apical over basal dendritic branch numbers (*n* = 14 BT neurons and 13 ST neurons, *P* = 0.003, Wilcoxon rank-sum test). **e** Comparison of the depth of the soma from the pial surface of BT and ST neurons (*n* = 14 BT neurons and 13 ST neurons, *P* = 0.021, two-sided Wilcoxon rank-sum test). **f** Dendritic density heatmap of BT and ST

neurons aligned to the pia (white circle show somata positions). **g** Examples of long-range projection patterns in S1 after the local injection in POm, VPM and M1 of an AAV vector expressing ChR2-YFP. **h** Fluorescence intensity profiles of POm, VPM, M1 and S2 long-range projections in S1 (each profile is an average from 3 mice and 2 slices per mouse). **i** Dendritic density profiles for BT and ST neurons across cortical layers. **j** Distributions of dot products for each long-range input profiles and the dendritic density of each BT and ST neurons (*n* = 14 BT neurons and 13 ST neurons, for POm, *P* = 0.002; for VPM, *P* = 0.03; for M1, *P* = $7.7 \times 10^{-5}$; for S2, *P* = $7.7 \times 10^{-5}$; two-sided Wilcoxon rank-sum test). For bar graphs and line profiles, error bars and shaded areas are s.e.m. For boxplots, the central line indicates the median, the box represents the interquartile range, the whiskers extend to the most extreme data points not considered outliers, and crosses indicate outliers. Source data are provided as a Source Data file.

monosynaptic inputs from POm. Therefore, the VPM-evoked PSPs in BT neurons were likely the result of polysynaptic circuit motifs.

To further investigate the different levels of thalamocortical input to the two groups of neurons, we utilized the anterograde trans-synaptic labeling properties of AAV1[35]. AAV1-mCaMKIIα-iCre-WPRE-hGHp(A) was injected in either the POm or VPM, and AAV2-hSyn-DIO-eGFP in S1 (Supplementary Fig. 4a). Owing to the trans-synaptic transport of the AAV1-Cre vector, GFP expression was driven in the neurons that held synaptic connections with thalamocortical axons. AAV1-Cre from the POm drove GFP expression predominantly in cortical L2/3 and L5 neurons, whereas from the VPM it labeled cells in L4 and L2/3 (Supplementary Fig. 4b). The L2/3 neurons that were targeted by POm afferents were on average located closer to the pial surface than those targeted by VPM afferents (Supplementary Fig. 4c). Since BT neurons tend to be positioned more superficially in the cortex as compared to ST neurons (Fig. 1e), these observations suggest that POm axons target

predominantly BT neurons, whereas VPM axons are biased to ST neurons. This corroborates our electrophysiological findings showing that BT neurons received on average stronger synaptic input from POm, and ST neurons stronger input from VPM (Fig. 2b).

## Inputs from POm afferents combined with other long-range synaptic inputs selectively induce NMDAR-dependent responses in BT neurons

Cortical L2/3 pyramidal neurons can integrate dendritic inputs in a supralinear manner, mediated by NMDARs, also called NMDA spikes[23,33,36–39]. These events are facilitated under depolarized conditions, when synapses are clustered, or when synapses harbor signaling mechanisms that strongly interact with one another[40–43]. L2/3 neurons have been shown to produce NMDA spikes upon sensory stimulation which may depend on inputs from POm and other afferents in L1[23,39]. Therefore, we sought to investigate whether the stimulation of different

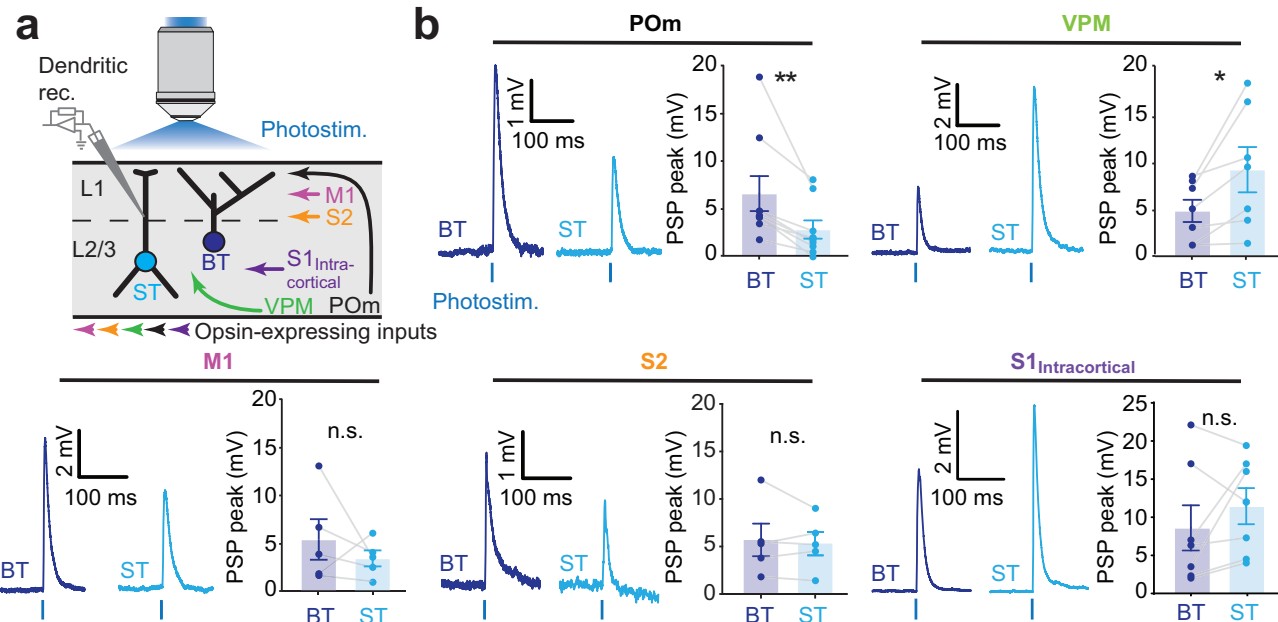

**Fig. 2 | Distinct input levels from different types of afferents on BT and ST neurons in S1. a** Experimental design that consisted in dendritic recordings of PSPs in BT and ST neurons upon selective photostimulation of various populations of afferent inputs. To account for the variability in expression levels of the opsin, a paired comparison between a BT and a ST neuron was performed within each brain slice. **b** Example traces are shown on the left and averaged PSP peaks were calculated for each recorded pair on the right. BT neurons exhibited larger responses than ST neurons when POm afferents were stimulated ($n = 9$ pairs; BT: $6.5 \pm 1.8$ mV; ST: $2.8 \pm 0.9$ mV; $P = 0.004$, two-sided Wilcoxon signed-rank test). However, ST neurons had significantly larger responses than BT neurons when VPM afferents were stimulated ($n = 7$ pairs; BT: $4.8 \pm 1.2$ mV; ST: $9.3 \pm 2.4$ mV; $P = 0.0154$, two-sided Wilcoxon signed-rank test). No significant differences were observed when M1, S2 and S1$_{intracortical}$ inputs were stimulated (for M1, $n = 5$ pairs; BT: $5.3 \pm 2.3$ mV; ST: $3.4 \pm 3.7$ mV; $P = 0.43$; for S2, $n = 5$ pairs; BT: $5.7 \pm 1.7$ mV; ST: $5.3 \pm 1.2$ mV; $P = 0.94$, for S1$_{intracortical}$, $n = 7$ pairs; BT: $7.7 \pm 3.3$ mV; ST: $12.0 \pm 3.4$ mV; $P = 0.2$, two-sided Wilcoxon signed-rank tests). Error bars, s.e.m. Source data are provided as a Source Data file.

combinations of thalamocortical and corticocortical afferents have distinct propensities to produce NMDA spikes in BT and ST neurons. We expressed ChR2 and ChrimsonR in various pairs of putative presynaptic afferents and then performed dendritic recordings from either class in brain slices while photostimulating two afferents simultaneously (Fig. 3a). We used light intensities and wavelengths that generated action potentials in the opsin-expressing neurons, but avoided cross-contamination between the two light channels (Supplementary Fig. 5, see ref. 44). The recorded dendrites were held at −55 mV to facilitate the generation of NMDA spikes. We first simultaneously photostimulated POm and M1 afferents. In BT cells this evoked seemingly two types of PSPs, characterized by smaller and larger amplitudes (Fig. 3b). The large-amplitude PSPs were prevented upon perfusion of the NMDAR antagonist APV, but the smaller amplitude PSPs remained unaffected. The large-amplitude PSPs were also prevented when NMDAR opening was precluded by holding cells at hyperpolarized potentials (~ −100 mV) (Supplementary Fig. 6a). Together, this confirms that the larger PSPs included NMDAR-mediated conductance and can be classified as NMDA spikes (Fig. 3b). NMDA spikes were also observed in ST neurons upon stimulation of VPM and S1$_{intracortical}$ afferents (Supplementary Fig. 6b). The occurrence of NMDA spikes was dependent on the combination of stimulated inputs rather than the stimulation strength or the initial PSP size (Supplementary Fig. 7).

To assess the efficacy by which various afferent pairs evoked NMDA spikes, we first determined whether the distribution of the evoked PSP event sizes was bimodal (see Methods, Fig. 3b, and Supplementary Fig. 6). Then, for each combination of inputs and each cell type, $k$-means clustering was used to separate the NMDA spikes from the regular PSPs, from which the fraction of trials with NMDA spikes as well as their total strength (fraction multiplied by amplitude) were computed. This analysis revealed that co-stimulation of POm and M1 afferents had a higher capacity to evoke NMDA spikes in BT neurons as

compared to ST neurons, as well as compared to the other tested combinations of putative inputs (Fig. 3c). Most BT neurons also displayed NMDA spikes when POm and VPM afferents were co-stimulated. This is intriguing since we could not detect distinct monosynaptic inputs coming from VPM afferents onto BT neurons (Supplementary Fig. 3). It implies that the stimulation of POm afferents combined with VPM-mediated activation of local excitatory circuits such as from L4 and ST neurons, can generate NMDA spikes. The generation of NMDA spikes in ST neurons was most pronounced when VPM afferents and S1$_{intracortical}$ circuits were co-stimulated (Fig. 3c). These data indicate that NMDA spikes can be evoked in L2/3 pyramidal neurons upon combined stimulation of long-range synaptic input. This bears similarities to the NMDA spikes that have been reported in L4 upon combined thalamocortical and intracortical stimulation[38], and the supralinear potentials in L2/3 pyramidal neurons upon sensory stimulation[23,39]. Our data show that the occurrence of NMDA spikes in L2/3 pyramidal neurons is cell-type dependent and relies on the activation of specific combinations of inputs.

## Inputs from POm afferents evoke delayed sustained dendritic potentials (DSDPs) in BT cell dendrites

While we assessed the functional connectivity of long-range inputs onto L2/3 pyramidal neurons, we observed that photostimulation of POm afferents using a train of 5 pulses (8 Hz) evoked sustained plateau-like depolarizations that followed the 5 short-latency PSPs with long and variable delays (Fig. 4a). Such events were virtually absent upon stimulation of other afferents.

To better characterize these events, we compared pairs of inputs on BT and ST neurons from ChR2 and ChrimsonR expressing afferents, which were independently tested at least 10 times by interleaving the two opsin photostimuli every 10 s. Dendritic recordings from BT neurons systematically displayed long-lasting depolarizing potentials

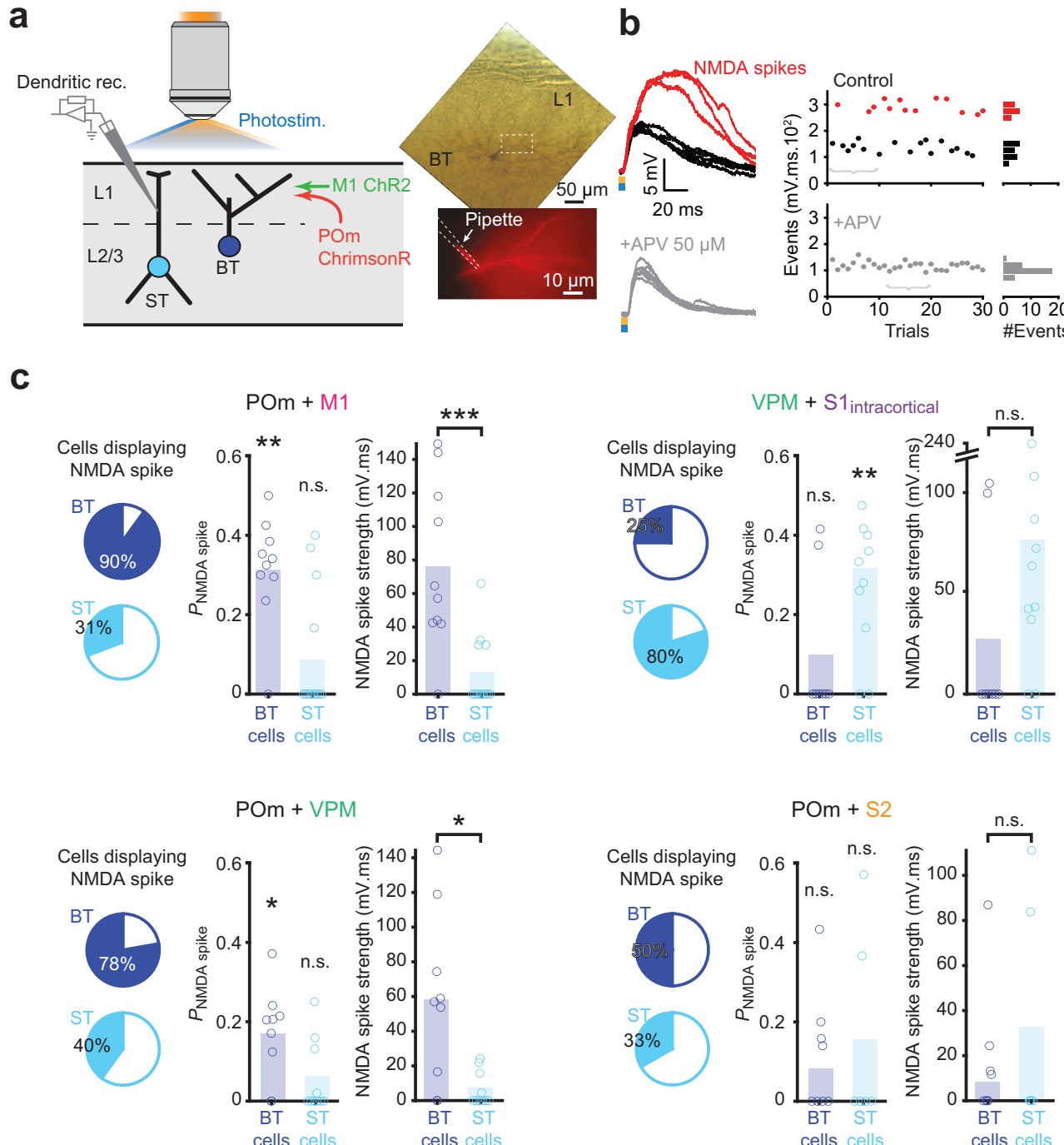

**Fig. 3 | Cell-type and input specific generation of NMDA spikes in L2/3 neurons. a** Experimental design for testing the integration of two inputs that converge onto BT and ST neurons in S1. The two inputs expressed different opsins (ChR2 and ChrimsonR) by local injection of AAV vectors. In this example, ChrimsonR was expressed in POm and ChR2 in M1. To ensure accurate recording of distal events, patch-clamp recordings were performed on the apical dendrites of BT or ST neurons. **b** Examples of events evoked by the selective co-stimulation of POm and M1 afferent inputs to a BT neuron (top). The stimulation either elicited regular PSPs (black traces) or NMDA spikes (red traces). The bath application of 50 μM APV prevented the generation of NMDA spikes (bottom). The distribution of the size of the events, from this example, showing that NMDA spikes can be easily segregated from regular PSPs (right; braces indicate the position of the displayed example traces on the left). **c** Occurrence of NMDA spikes for four different input combinations. For each of them, the pie charts indicate the percentage of BT and ST cells that displayed NMDA spikes at least once during the recording period (left). For each of the cell types, the NMDA spike probability per trial was compared to the null hypothesis of a zero probability (middle, for POm + M1: BT cells, $n = 10$, $P = 3.9 \times 10^{-3}$; ST cells, $n = 13$, $P = 0.12$; for VPM + S1$_{intracortical}$: BT cells, $n = 8$, $P = 0.5$; ST cells, $n = 10$, $P = 7.8 \times 10^{-3}$; for POm + VPM: BT cells, $n = 9$, $P = 0.02$; ST cells, $n = 10$, $P = 0.12$; for POm + S2: BT cells, $n = 8$, $P = 0.12$; ST cells, $n = 6$, $P = 0.5$; two-sided Wilcoxon rank sum tests). The NMDA spike strength between BT and ST groups was compared (right, for POm + M1: $P = 8.7 \times 10^{-3}$; for VPM + S1$_{intracortical}$: $P = 0.09$; for POm + VPM: BT cells, $P = 0.054$; for POm + S2: BT cells, $P = 0.93$; same ns as for the fraction of trials, two-sided Wilcoxon sign rank tests). Source data are provided as a Source Data file.

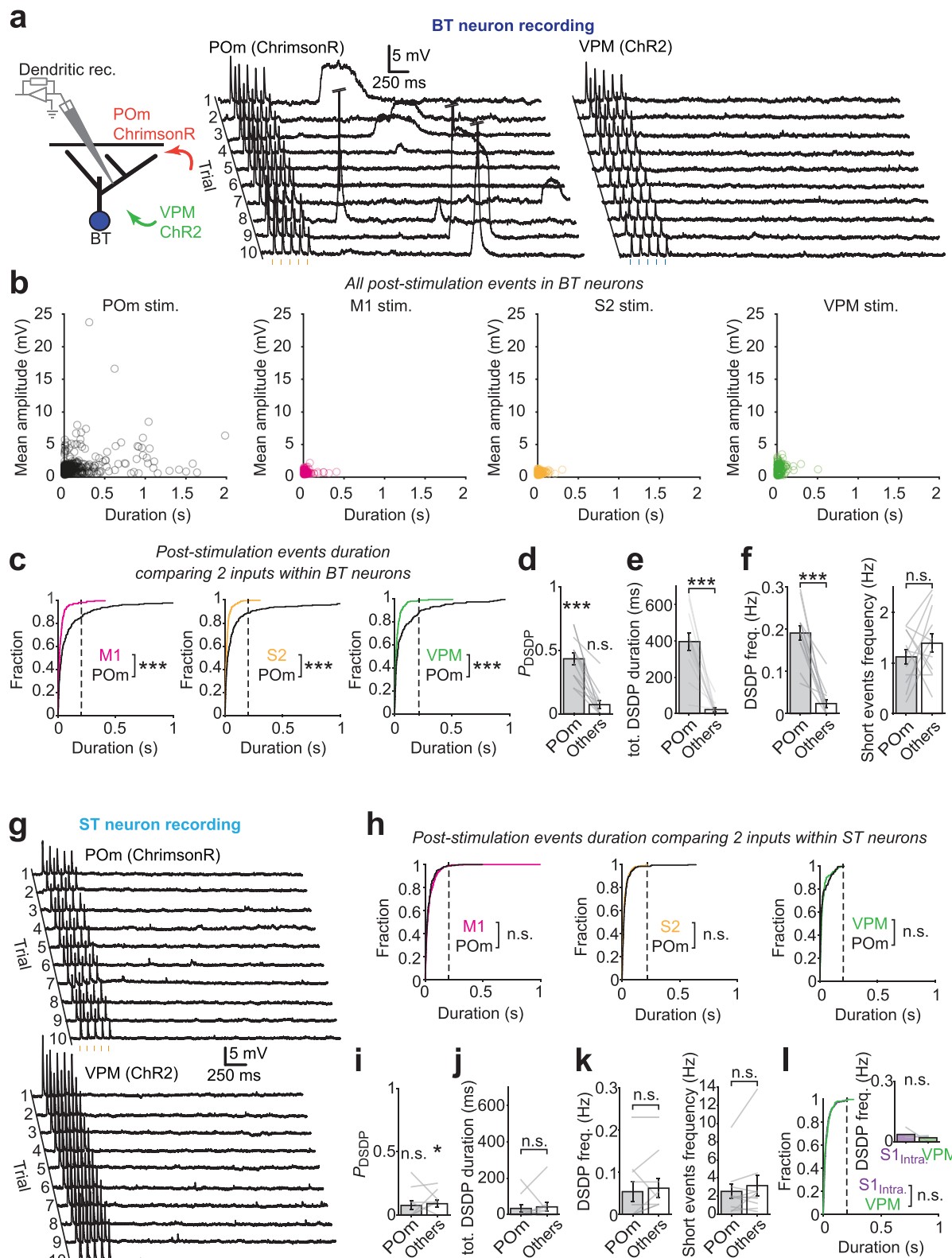

following photostimulation of POm but not of VPM (Fig. 4a). We did not observe this in ST neurons (Fig. 4g). To quantitatively assess these events, we designed a filter to detect any depolarization that occurred after the 5 stimuli and then measured their durations and amplitudes. Stimulation of POm afferents often evoked large-amplitude and long-lasting events, which were not seen after stimulation of M1, S2, and VPM (Fig. 4b). The cumulative distribution of the event durations indicates

that inputs from POm consistently produced longer-lasting events as compared to the other inputs (Fig. 4c). A similar analysis of the recordings from ST neurons did not reveal differences in the cumulative distribution of event durations between the POm simulation and any other inputs (Fig. 4h).

These long-lasting depolarizations were clearly distinct from regular PSPs and NMDA spikes. Their average duration, rise time, and

**Fig. 4 | POm activation induces long-lasting and delayed DSDPs in BT neurons.**
**a** Example traces of a dendritic recording from a BT neuron when POm afferent inputs expressing ChrimsonR and VPM afferent inputs expressing ChR2 were photostimulated independently. In both cases, the five pulses of light elicited PSPs. Long-lasting DSDPs were regularly observed after POm afferents were stimulated. These events occurred with a highly variable delay and were not observed following VPM afferent stimulation. **b** Scatter plots showing the duration and mean amplitude of all post-stimulation events that were automatically detected in BT neurons. While only short duration events with small amplitudes (corresponding to spontaneous PSPs) were detected following the stimulation of M1, S2 or VPM afferents, long duration events of variable amplitude were detected following POm stimulation. **c** Cumulative fractions of the post-stimulation event durations for POm and other afferent inputs (M1, S2 and VPM) to BT neurons (left, M1 vs. POm: $P = 1.7 \times 10^{-7}$, S2 vs. POm: $P = 1.6 \times 10^{-10}$, VPM vs. POm: $P = 5.5 \times 10^{-5}$, two-sided Koglomorov-Smirnov tests). **d** Probabilities (per trial) of detecting at least one DSDP following photostimulation of POm vs M1, S2 and VPM afferents, tested against the zero probability ($n = 12$ neurons, for POm: $P = 4.8 \times 10^{-4}$; for others: $P = 0.06$, two-sided Wilcoxon sign-rank tests). **e** Total duration of DSDP events per trial in BT neurons for POm compared to other

inputs ($n = 12$ neurons, $P = 2.4 \times 10^{-5}$; two-sided Wilcoxon sign-rank test) **f** Left, DSDPs frequency in BT neurons for POm compared to other inputs ($n = 12$ neurons, $P = 4.8 \times 10^{-4}$; two-sided Wilcoxon sign-rank test). Right, same comparison for short events frequency ($P = 0.37$; two-sided Wilcoxon sign-rank test). **g** Example traces of the dendritic recording of an ST neuron after independent photostimulation of POm and VPM afferent inputs. No DSDPs were observed when POm or VPM afferents were stimulated. **h** Cumulative fractions of the post-stimulation event durations for various pairs of inputs in ST neurons (left, M1 vs. POm: $P = 0.58$, S2 vs. POm: $P = 0.35$, VPM vs. POm: $P = 0.052$, Koglomorov-Smirnov tests). **i** Same analysis as (**d**) for ST neurons ($n = 10$, for POm, $P = 0.12$; for other inputs: $P = 0.03$; two-sided Wilcoxon sign-rank test). **j** Same analysis as (**e**) for ST neurons ($n = 10$, $P = 0.77$, two-sided Wilcoxon sign-rank test). **k** Same analysis as (**f**) for ST neurons ($n = 10$, for DSDP frequency, $P = 0.62$; for short events frequency, $P = 0.32$, two-sided Wilcoxon sign-rank test). **l** Cumulative fractions of the post-stimulation event durations for S1$_{intracortical}$ and VPM inputs in ST neurons ($P = 0.054$, Koglomorov–Smirnov tests). Inset shows the DSDPs frequency in ST neurons for this pair of inputs ($n = 4$, $P = 0.44$; two-sided Wilcoxon sign-rank test). Error bars, s.e.m. Source data are provided as a Source Data file.

amplitudes were significantly different from the NMDA spikes seen upon combined stimulation of inputs (Supplementary Fig. 8). However, they were characterized by a certain degree of variability in their onset, amplitude, and duration (Supplementary Fig. 9). Together, this indicates that these events were very unlikely to represent spontaneous AMPAR-mediated PSPs or NMDA spikes. To distinguish them from NMDA spikes, we termed them DSDPs for the remainder of the paper.

The probability of detecting DSDPs (>200 ms) in BT neurons after stimulation of POm afferents was significantly higher than after stimulation of any of the other inputs (Fig. 4d). Even though they sporadically arose after stimulation of the other inputs, their probability was not significantly above zero (Fig. 4d). Thus, the occurrence of DSDPs, as measured by their probability per trial, their total duration per trial or overall frequency per cell, appeared exclusively associated with stimulation of POm afferents (Fig. 4e, f). The increase in DSDP frequency was independent of the short duration event frequency, which was not different between POm and other afferent stimulation (Fig. 4f). Stimulation of POm afferents did not increase their probability in ST neurons (Fig. 4i) as compared to other afferent stimuli, and no difference was found when comparing the total duration and frequency of DSDPs (Fig. 4j, k), as well as the frequency of short-lasting events (Fig. 4k). Moreover, when comparing the VPM and S1$_{intracortical}$, we almost exclusively found short-lasting events and the measured frequencies of DSDPs were close to zero (Fig. 4l). In addition, BT neurons in M1 did not produce any DSDPs upon photostimulation of POm afferents (Supplementary Fig. 10). Together, these results indicate that DSDPs predominantly occurred in S1 BT neurons and were selectively associated with the stimulation of POm inputs.

## DSDPs depend on the closing of leak K$^+$ channels
Since the DSDPs were so distinct from any of the other events, we hypothesized that they were associated with the opening or closing of ion channels other than the typical synaptic receptors. To investigate the conductance that was mediating the DSDPs, we recorded from BT neurons while photostimulating POm afferents and holding the membrane potential at −50, −90, and −120 mV (Fig. 5a). Whereas the DSDPs were depolarizing at −50 mV, they nearly disappeared at −90 mV and became hyperpolarizing at −120 mV. We inferred that the reversal potential of these events was around −94 mV (Fig. 5b). This is consistent with the potassium conductance, which under our experimental conditions was estimated to be around −106 mV (see Methods). Considering that DSDPs were detected as depolarizing at resting membrane potentials, we deemed it unlikely that they were mediated by the opening of hyperpolarizing and voltage-dependent potassium channels. Instead, we hypothesized that they were mediated by leak

potassium channels that regulate resting membrane potentials, the majority of which is formed by the K2P channel family[45]. Under our experimental conditions, the DSDPs would thus reflect the transient closing of the K2P channels. To test this, we performed dendritic recordings of BT neurons, while photostimulating POm afferents. We measured the DSDP frequency before and after bath application of a broad-spectrum cocktail of K2P channels blockers[45] (Fig. 5c). Consistent with the hypothesis, the blocking of these channels significantly reduced the frequency of DSDPs (Fig. 5c, d). In conclusion, the perfusion of a broad-spectrum cocktail of blockers alone, without POm stimulation, induced a transient depolarization accompanied by an increase in EPSP frequency, consistent with elevated firing activity in connected neurons (Supplementary Fig. 11). To narrow down which K2P channel subtypes could be involved, we tested the effect of more specific antagonists in a different set of experiments. Blocking TASK or TREK channels by bath application of A1899 or fluoxetine significantly reduced the DSDPs frequency (Fig. 5e), whereas the blocking of THIK-1 channels by IBMX did not affect them (Fig. 5f). Finally, the blocking of K2P channels that include TASK and TREK channels increased the membrane resistance of the recorded neurons (Fig. 5g), suggesting that the effects were cell autonomous.

## DSDPs in POm to BT synaptic inputs are mediated by post-synaptic group I mGluRs
We next sought to investigate how the activation of POm to BT synaptic inputs leads to the blocking of TASK/TREK channels. TASK and TREK channels have been shown to be modulated by G protein-coupled receptor (GPCR) signaling pathways[46,47]. The mGluRIs have been shown to induce delayed and long-lasting depolarizing events resembling the ones observed in our recordings[21,22,48]. Therefore, we hypothesized that the activation of POm to BT synaptic inputs triggers mGluRI-signaling which subsequently mediates the transient closing of TASK or TREK channels. To test this, we performed another set of dendritic recordings on BT neurons and measured the frequency of DSDPs following POm stimulation before and after bath perfusion of specific and generic mGluRI blockers, LY367385 and MCPG respectively (Fig. 6a, b). Blocking mGluRIs significantly reduced the frequency of DSDPs, similar to the effect of blocking of K2P channels (Fig. 6b). Additionally, perfusion of MPEP, a selective non-competitive mGluR5 antagonist, did not reduce the number of DSDPs, suggesting that mGluR1 is the predominant receptor involved in triggering these events (Fig. 6d, e). To confirm that DSDPs were mediated by post-synaptic mGluRI, we added GDP-β-S to the intracellular solution (Fig. 6d). GDP-β-S is a non-hydrolyzable analog of GDP that internally blocks G-protein activity[17,49]. Whereas photostimulation of POm still

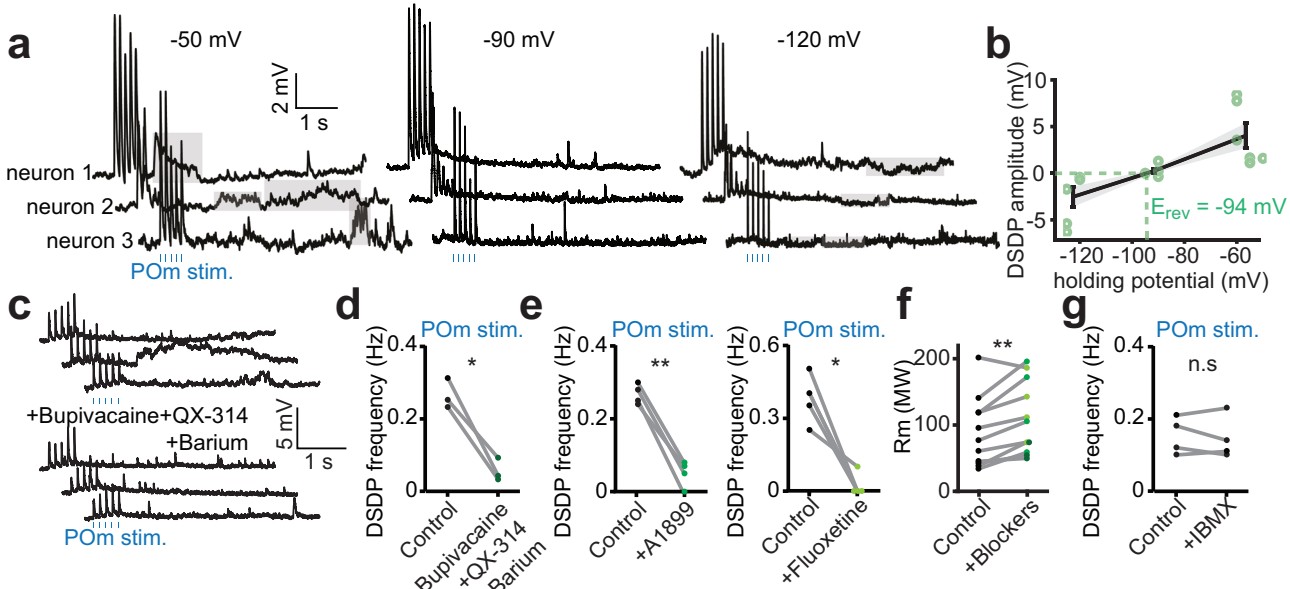

**Fig. 5 | POm-mediated DSDPs are due to the closing of K2P channels. a** Dendritic recordings of a BT neuron held at different holding potentials during the photo-stimulation of POm afferents (5 pulses of 5 ms at 8 Hz, every 10 s). At a holding potential of −60 mV, DSDPs appear as long-lasting depolarizing events (gray patches). At a holding potential of −125 mV the DSDPs became hyperpolarizing events. **b** DSDP amplitude evoked by the stimulation of POm as a function of the holding potential ($n$ = 6 neurons, from 5 mice). The reversing potential of these events was measured at −94 mV consistent with a potassium conductance (see Methods). Error bars indicate s.e.m. **c** Dendritic recordings of a BT neuron during the stimulation of POm afferents (5 pulses of 1 ms at 8 Hz, every 10 s) before and after bath application of bupivacaine (1 mM), QX-314 (1 mM) and barium (1 mM). No DSDPs could be observed in the presence of this non-specific blockade of K2P channels. **d** DSDPs

frequency was significantly reduced in the presence of non-specific K2P channels blockers ($n$ = 3, from 3 mice, $P$ = 0.023, two-sided paired t-test). **e** The selective blockade of TASK or TREK channels using A1899 (100 nM) or fluoxetine (100 μM) respectively, largely prevented the generation of DSDPs (for A1899, $n$ = 4, from 3 mice, $P$ = 0.001; for fluoxetine, $n$ = 4, from 3 mice, $P$ = 0.018, two-sided paired t-tests). **f** Altogether, the K2P blockers used in (**d**, **e**) significantly increase the membrane resistance of the recorded dendrites (control: 88.5 ± 15.9 MΩ, with blockers: 111.3 ± 16.6 MΩ, $n$ = 11, from 3 mice, $P$ = 0.004, two-sided paired t-test). **g** Blocking the THIK channel family with IBMX (1 mM) did not produce any change in the frequency of DSDPs ($n$ = 4, from 3 mice, $P$ = 0.62, two-sided paired t-test). Source data are provided as a Source Data file.

induced DSDPs immediately after break-in with the patch electrode, the events were largely abolished within approximately 3 min (Fig. 6f, g), which is consistent with the dialysis kinetics of GDP-β-S[50]. Overall, the presence of mGluRIs or G-protein blockers slightly decreased the membrane resistance of all recorded neurons (Fig. 6c, h). Conversely, the bath application of the mGluRI agonist DHPG increased both the frequency and the amplitude of DSDPs triggered by POm stimulation, along with an increase in the input resistance (Supplementary Fig. 12a–c). This effect was absent in ST neurons (Supplementary Fig. 12d, e). To further investigate the possible selective expression of mGluRI in BT neurons, we performed single-cell mRNA-sequencing analysis on patched BT and ST neurons. The results indicate a trend toward higher transcript levels for mGluRI genes in BT neurons compared to ST neurons (Supplementary Fig. 13a, b). This is congruent with the Allen Brain Atlas (mouse brain ISH data), which shows that transcripts of mGluRI genes are present in L2/3 neurons, with a bias to the upper segment of L2/3[51] (Supplementary Fig. 13c). Additionally, integrating the expression profiles into the Allen Brain Cell Types RNA-Seq database[52–54], we observed that BT and ST neurons tend to separate towards distinct canonical transcriptomic L2/3 pyramidal cell subtypes (Supplementary Fig. 13d). Taken together, these experiments indicate that DSDPs are mediated by the signaling of postsynaptic mGluRIs at the POm to BT synaptic inputs, which is supported by the expression profiles of mGluRI transcripts in BT neurons and upper layer L2/3 neurons in general.

## Modulation of mGluRIs alters movement-associated spiking of L2/3 neurons in vivo

mGluRI-mediated DSDPs and the associated increase in input resistance may represent a mechanism for increasing the gain of

concomitant synaptic inputs. Indeed, occasionally action potentials were superimposed on the DSDPs (Fig. 4a). To further investigate this, we performed dendritic recordings of BT neurons while bath applying TBOA, a glutamate-reuptake inhibitor that prolongs the presence of ambient glutamate in (and around) the synapse[55,56]. Under these conditions, the photostimulation of POm afferents increased the occurrence of action potentials that were superimposed on the DSDPs (Supplementary Fig. 14a, b). This effect was absent upon stimulation of M1 afferents, despite the amplifying effect of TBOA on the evoked PSPs (Supplementary Fig. 14c). This suggests that the DSDPs are the leading cause for the increased occurrence of action potentials triggered by NMDAR-mediated events, as previously reported[57,58]. Affirmatively, the high amplitude DSDPs and spikes disappeared when NMDARs were blocked by adding APV to the bath, but this did not impact the duration of the DSDPs (Supplementary Fig. 14a, b). These could only be removed by an additional inhibition of mGluRI (Supplementary Fig. 14a, b). Altogether, the data indicate that under a prolonged presence of glutamate – a phenomenon that may mimic conditions as they occur during bursting activity, mGluRI-mediated DSDPs may promote the generation of somatic action potentials.

These observations incited us to explore how the modulation of mGluRI affects the activity of cortical neurons in vivo. Mice actively use their whiskers to sense their environment, which consists of volitional movements that are in part initiated by activity in motor cortices, among which M1[59]. Neurons in the vibrissal area of M1 encode whisking parameters during active sensing behavior, and this activity is subsequently transmitted back to L1 of S1[60]. Based on our observations, we argued that POm-mediated activation of mGluRI could selectively increase the gain of incoming motor signals from M1 onto BT neurons

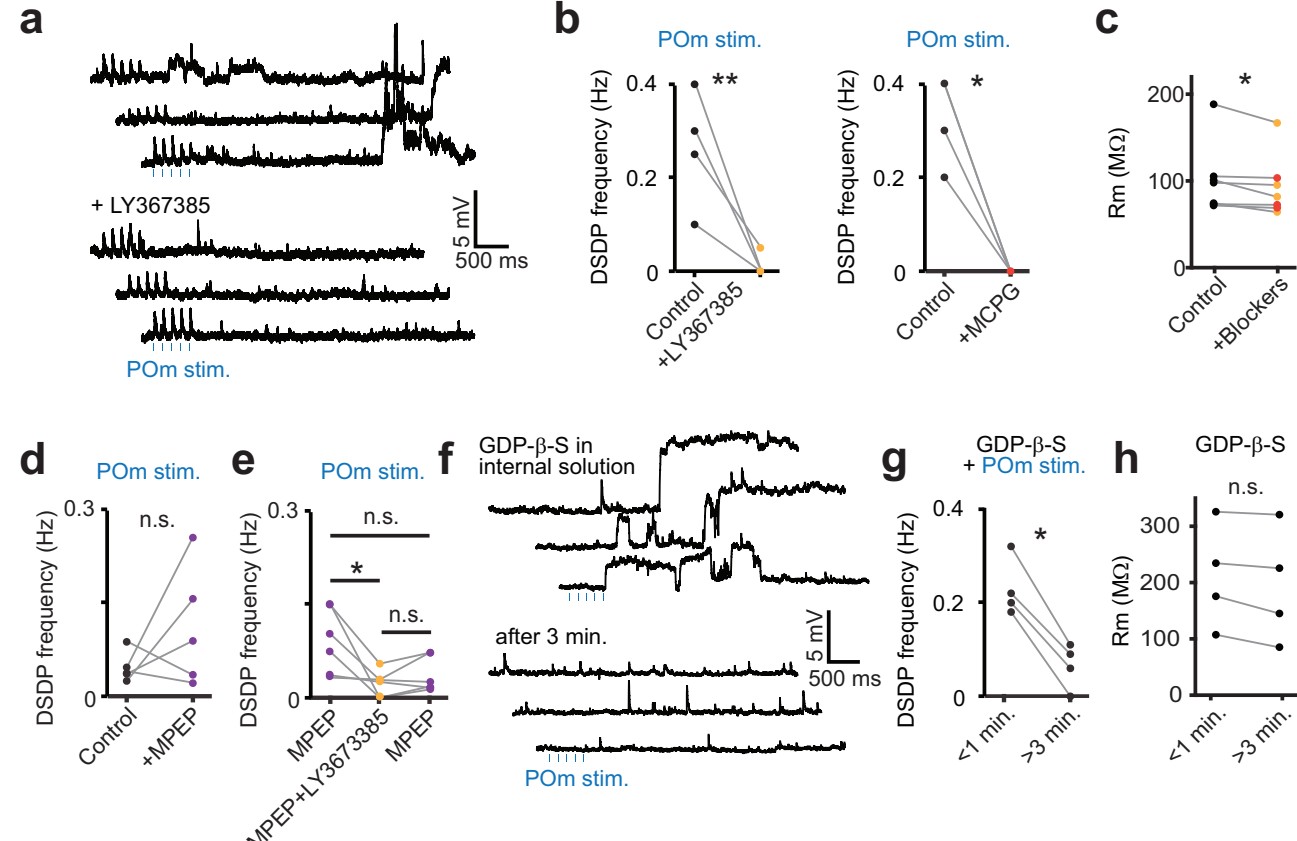

**Fig. 6 | POm-mediated DSDPs are mediated by the activation of mGluRIs.**
**a** Dendritic recordings of a BT neuron during the stimulation of POm before and after the bath application of LY367385 (50 μM), a selective mGluR1 blocker. LY367385 prevented the generation of DSDPs. **b** The DSDP frequency was mostly prevented in the presence of LY367385 (50 μM; *n* = 4, from 4 mice, *P* = 0.003, two-sided paired t-test) as well as in the presence of the generic mGluRI blocker MCPG (500 μM; *n* = 3, from 3 mice, *P* = 0.03, two-sided paired t-test). **c** Together, the mGluRI blockers LY367385 and MCPG significantly decrease the membrane resistance of the recorded dendrites (control: 101.4 ± 15.4 MΩ, with blockers: 93.0 ± 13.4 MΩ, *n* = 7, from 4 mice, *P* = 0.04, two-sided paired t-test). **d** Dendritic recordings of BT neurons in the presence of MPEP, a selective mGluR5 blocker, did not significantly change the DSDP frequency following POm afferent photo-stimulations (10 μM; *n* = 5, from 5 mice, *P* = 0.25, two-sided paired t-test). **e** In

another set of recordings, LY367385 was subsequently added to the MPEP containing ACSF solution which significantly reduced the DSPD frequency (*n* = 6, from 4 mice, *P* = 0.029, two-sided paired t-test). DSDP frequency did not significantly recover after LY367385 washout (*P* = 0.14; MPEP pre vs MPEP post, *P* = 0.08; two-sided paired t-test). **f** Dendritic recording of a BT neuron during the stimulation of POm afferents in the presence of GDP-β-S (1 mM), a G-protein activity blocker, in the intracellular solution. Within the first minute after break-in, DSDPs could be observed but then disappeared after 3 min, consistent with the dialysis of the drug. **g** The frequency of DSDPs was largely reduced after intracellular dialysis of GDP-β-S (*n* = 4, from 4 mice, *P* = 0.003, two-sided paired t-test). **h** The dialysis of GDP-β-S did not significantly reduce the membrane resistance of the dendrite (control: 210.8 ± 46.3 MΩ, with blockers: 194.0 ± 51.0 MΩ, *n* = 4, from 4 mice, *P* = 0.066, two-sided paired t-test). Source data are provided as a Source Data file.

by enhancing their excitability. Therefore, we hypothesized that the activation of mGluRI would increase the propensity for active whisking to induce somatic spikes in S1 L2/3 pyramidal neurons. To investigate this, we performed in vivo 2-photon laser scanning microscopy to image calcium (Ca²⁺) signals in S1 L2/3 neurons expressing GCaMP6s. Ca²⁺ signals were recorded for 10 min before and after modulating mGluRIs using the local infusion of the agonist DHPG or the antagonist MCPG (Fig. 7). Using a piezo-driven microscope objective, we imaged near-simultaneously the upper and lower L2/3 neurons (at −100 and −300 μm distance from the pia, respectively; Fig. 7a). For the analysis, we assumed that the population of BT neurons is enriched in upper L2/3 while the location of ST neurons is more biased towards deeper L2/3 (Fig. 1e). We first compared the level of the overall activity of individual neurons in upper and lower L2/3 before and after DHPG (Fig. 7b, d, e) or MCPG infusion (Fig. 7c, f, g). DHPG increased the overall activity of L2/3 neurons but more so in upper L2/3 (Fig. 7d, e). MCPG modestly increased the overall activity, but substantially suppressed activity of a subset of upper L2/3 neurons (Fig. 7f, g). We tracked snout movements as a proxy of whisking[61] using DeepLabCut[62]. A random forests

decoding algorithm was trained to predict snout movements from the activity of individual neurons (Fig. 7h, i). The correlation coefficient between the predicted and actual movements, which we defined as the prediction power (PP), was calculated for each neuron before and after infusion of the drugs on the entire recording periods (Fig. 7j−m). While the average PP of upper and lower L2/3 neurons was similar in baseline conditions, we found that DHPG significantly increased the PP for upper but not for lower L2/3 (Fig. 7j, k). Conversely, MCPG significantly decreased the PP of upper but not of lower L2/3 neurons (Fig. 7l, m). We controlled that these changes could not be attributed to an alteration of the noise level during the recordings due to the drug injections (Supplementary Fig. 15). In addition, the PP of upper L2/3 neurons under control conditions showed a modest correlation with the magnitude of the increase in activity under DHPG and a strong correlation with the reduction in activity under MCPG. This was not observed for lower L2/3 neurons (Supplementary Fig. 16). Since sub-threshold depolarizations and changes in membrane excitability cannot be directly captured by two-photon calcium imaging, we performed additional in vivo whole-cell patch-clamp recordings in

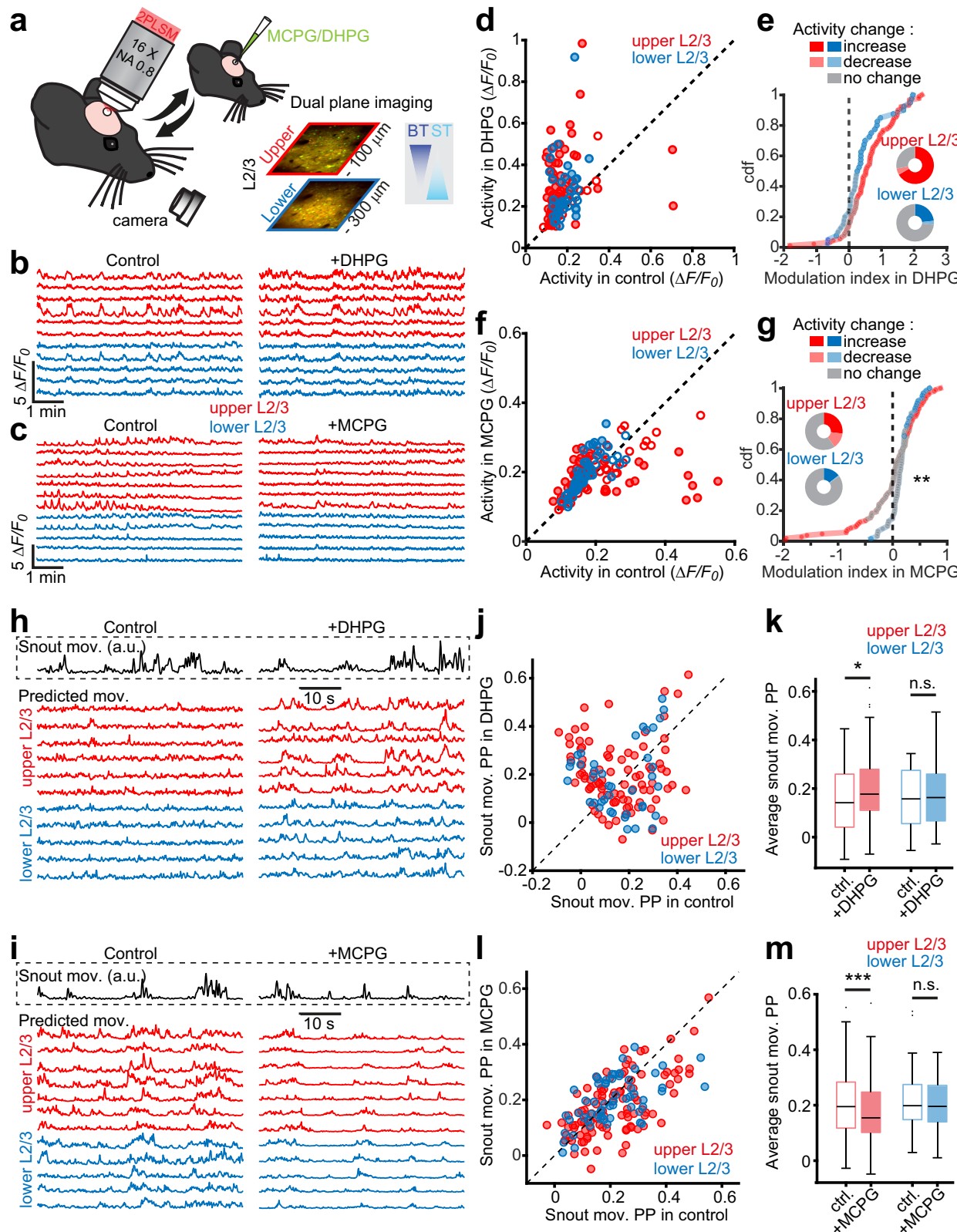

anesthetized mice to assess the effect of mGluRI modulation on superficial layer 2/3 pyramidal neurons (Supplementary Fig. 17). These recordings showed that mGluRI activation using DHPG significantly increased the intrinsic excitability of L2/3 pyramidal neurons, as indicated by its depolarizing effect on the resting membrane potential and its elevating effect on the input resistance. Together, these findings suggest that mGluRI signaling in a subset of upper L2/3 neurons

enhances their propensity to generate spikes during whisking, likely by boosting intrinsic excitability.

## Discussion

We showed that activation of thalamocortical projections to S1 L2/3 pyramidal neurons from the POm promotes the generation of NMDA spikes when combined with other inputs. POm thalamocortical inputs

**Fig. 7 | Bidirectional modulation of movement-related integration in upper L2/3 neurons by mGluRIs in vivo. a** In vivo 2-photon calcium imaging was performed before and after DHPG or MCPG injection through a silicone port in the barrel cortex. Upper and lower L2/3 neurons expressing GCaMP6s and mRuby2 were recorded quasi-simultaneously at −100 and −300 μm from the pial surface. **b** Example traces of upper (red) and lower (blue) L2/3 neurons before and after DHPG injection. **c** Same as (**b**) for MCPG injection. **d** Average normalized activity during baseline and after DHPG for upper (red, $n = 90$ neurons) and lower (blue, $n = 40$ neurons) L2/3 neurons (5 mice). Activity changes (closed circles) were defined by a permutation test ($P < 0.01$) and small effect size (Cohen's d > 0.2); open circles represent neurons that do not change their activity. **e** Cumulative distribution of modulation index (relative change after vs. before DHPG) for upper and lower L2/3 neurons ($P = 0.056$, two-sided Kolmogorov-Smirnov test). Pie charts show proportions of neurons increasing, decreasing, or not changing activity. **f** Same analysis as (**d**) after MCPG injection ($n = 104$ upper and 61 lower neurons,

from 3 mice). **g** Cumulative distribution and proportions of neurons showing activity changes after MCPG injection ($P = 0.005$, two-sided Kolmogorov-Smirnov test). Snout movement prediction by individual neurons assessed using a random forests model before and after DHPG (**h**) or MCPG (**i**). Predicted (colored traces) and actual (black trace) snout movements for neurons in (**b**, **c**) are shown. **j** Comparison of snout movement predictive performance (PP) at baseline and after DHPG injection ($n = 90$ upper, 40 lower neurons, from 5 mice). **k** Distribution of snout movement PP after DHPG: increased in upper ($P = 0.026$) but not lower L2/3 neurons ($P = 0.44$; two-sided paired t-tests). **l** Same analysis as (**j**) for MCPG injection ($n = 104$ upper and 61 lower neurons, from 3 mice). **m** Snout movement PP decreased for upper but not lower L2/3 neurons after MCPG ($P = 4 \times 10^{-4}$ for upper; $P = 0.46$ for lower; two-sided paired t-tests). Boxplots: center, median; box, interquartile range; whiskers, data range excluding outliers (crosses). Error bars, s.e.m. Source data are provided as a Source Data file.

also have the capacity to evoke DSDPs via a mGluRI-mediated modulation of K2P channels. These effects are selective for a subpopulation of pyramidal neurons with broad apical tufts which are predominantly located in L2. Both phenomena actively increase neuronal excitability, which may augment these neurons' propensity to trigger spikes. We found support for this mechanism in vivo by demonstrating that mGluRI signaling increases excitability and preferentially enhances movement-associated spiking of L2 pyramidal neurons in S1.

### POm thalamocortical projections preferentially connect with L2/3 BT neurons

Using an unbiased classification, we clustered L2/3 pyramidal neurons in two groups, one with broad and dense apical tufts, and another with slender tufts (Fig. 1). The morphological characteristics of these populations are very similar to previously reported BT and ST neurons[8,9]. Our experiments show that the extent and density of their dendrites were good indicators of the synaptic connectivity with the axons that overlapped with them (Fig. 2), which is in line with Peters' rule stating that synaptic connections are proportional to the axo-dendritic overlap[12]. Stimulation of POm, M1 and S2 axons all tended to evoke larger PSPs in BT as compared to ST neurons, although the S2 and M1 functional inputs were not significantly different between both cell types. In contrast, stimulation of VPM and intracortical circuits evoked larger PSPs in the ST neurons which have a higher proportion of basal dendrites. We verified that this relationship was also present at the level of monosynaptic inputs between these neurons and POm or VPM axons (Supplementary Fig. 3). We did not perform this experiment for S2 and M1 afferents. Therefore, we cannot exclude that the lack of discrimination between those inputs onto BT and ST neurons was due to abundant polysynaptic connections from local cortico-cortical circuitry that may be activated by these pathways. The relatively strong connectivity between L1 afferents and BT neurons is in line with the notion that many L2 neurons in S1 and L2/3 neurons in V1 bearing complex dendritic arbors have relatively large receptive fields, since many L1 inputs derive from long-range projections originating from various brain regions[7,25,63].

The high connectivity rates of POm afferents with L2/3 BT neurons stand in contrast with the relatively low connectivity rates of those afferents with the abundantly present branches from broad tufted L5b neuron apical dendrites in L1, and the high rates with less abundant slender L5a neurons[4]. Thus, POm afferents provide input selectively to L2 BT neurons and L5a pyramidal neurons, which are highly interconnected[1,3,16]. This suggests that together they constitute a paralemniscal cortical circuit motif with distinct functions[64]. It is also interesting to note that the high connectivity rates of L2 neurons (i.e., L2/3 BT neurons) with POm afferents are associated with higher-than-average levels of plasticity[25,64,65]. However, there is no indication that the BT neuronal subtypes should generally display higher levels of activity in vivo[9].

### Converging POm and M1 inputs on L2/3 BT neurons cooperate to generate NMDA spikes

NMDA spikes were readily generated in BT neurons when POm afferents were co-stimulated with inputs from M1 and VPM, but not upon co-stimulation of VPM and S1$_{intracortical}$ afferents (Fig. 3). Conversely, NMDA spikes were abundant in ST neurons upon co-stimulation of VPM and S1$_{intracortical}$ afferents but not when POm afferents were co-stimulated with those from M1 or VPM. Co-stimulation of POm and S2 afferents did not produce a significant number of NMDA spikes in either BT or ST neurons. The efficacy at which POm together with M1, and VPM together with S1$_{intracortical}$, evoked NMDA spikes in BT and ST neurons, respectively, indicates that their synapses bolster supralinear synaptic interactions. Well known parameters for such interactions are synaptic proximity, the caliber of the parent dendrite, synaptic receptor content, the temporal order of activation, and the levels of local inhibitory and neuromodulatory input[37]. Favoring conditions are indeed met for the synaptic inputs of some of the above afferents. POm and M1 afferents both are likely to have dense connections with the apical dendrites of BT pyramidal neurons in L1, and VPM and S1$_{intracortical}$ afferents may densely connect with basal dendrites of ST pyramidal neurons in L3. Thus, the occurrence of NMDA spikes may be related to the convergence of these inputs onto single dendritic domains, which aligns with models in which clustered inputs favor supralinear synaptic integration[66–70]. In this respect, the finding that co-stimulation of POm and VPM afferents also evoked NMDA spikes in BT neurons was surprising, since VPM axons had virtually no monosynaptic connections with them. The most likely explanation for this finding is that they were triggered by the activation of polysynaptic L4-to-L2/3 and L2/3-to-L2/3 circuits that are readily recruited by VPM inputs, and which are more broadly distributed along the dendrites. In addition to their convergence, another favoring condition for NMDA spikes is provided by the location of POm synaptic inputs on thin distal endings of the apical branches[71], which may increase their cooperativity with other inputs[42]. In support of this, NMDA-mediated synaptic Ca$^{2+}$ responses can readily be observed in distal dendritic branches[23,39]. In a similar vein, the generation of NMDA spikes in ST neurons could be explained by the projection of VPM afferents onto their thin basal dendrites which are also favorable for evoking dendritic spikes[72,73]. A similar supra-linear interaction between VPM and local inputs has been described for L4 granule cells[38]. Furthermore, co-stimulation of POm afferents with local ascending cortical circuitry has been shown to drive disinhibition of L2/3 neurons[26]. This could aid the generation of NMDA spikes, which are highly sensitive to dendritic inhibition[74]. Modeling studies indeed suggest that disinhibition could gate the generation of NMDA spikes evoked by clustered inputs[75]. L2 pyramidal neurons receive substantial inhibition and could thus be powerfully controlled by such a mechanism[76]. Lastly, the various inputs that we tested could harbor temporally different synaptic activation patterns, which have been shown to be critical for supra-linear synaptic integration[77]. Further

experiments are needed to reveal if such relationships exist between long-range synaptic inputs to apical dendrites in L1.

Even though NMDA spikes often remain below the main spiking threshold, they are important for facilitating spikes initiated at the axon initial segment. Dendritic NMDAR-mediated events in vivo have been correlated with increased somatic spiking probabilities[38,39,78]. Thus, POm afferents may modulate activity of BT neurons through facilitating the generation of NMDA spikes in collaboration with other inputs. Subthreshold NMDAR-events have also been shown to drive synaptic plasticity[23,43,79], and previous work from our laboratory indicates that in L2/3 neurons this depends on input from the POm. Combined with the current insights this suggests that POm-mediated plasticity might be an attribute of L2/3 BT neurons. It will be interesting to investigate whether this relates to aspects of sensory learning.

### Stimulation of POm afferents modulates K2P leak channels through mGluRI signaling

POm stimulation in vivo has been shown to evoke sustained depolarizations in L2/3 and L5 neurons in vivo, which in turn may prolong their sensory-evoked responses[23–25,27]. Reverberatory activity in L2 is likely evoked through a combination of L5-to-L2 and direct POm-to-L2 inputs[25,64]. POm-mediated NMDA spikes may in part explain such effects, but additional biophysical mechanisms for these phenomena are likely at play given the long-lasting effects seen in the studies above (Supplementary Fig. 10). Upon a burst of POm stimuli we observed abrupt depolarizations specifically in BT neurons, which were temporally unlocked from the stimulus onset and varied in duration (Fig. 4). These potentials were very distinct from PSPs and synaptic NMDA spikes (Supplementary Fig. 8). Instead, they had similar kinetics to previously reported local glutamate-evoked dendritic plateau potentials that were sustained at 10 to 20 mV for 200–500 ms[58,80]. In these studies, the potentials propagated to the cell body where they could trigger a burst of action potentials, an effect that we did not further investigate in brain slices. Surprisingly, in our experiments, we found that DSDPs were mediated by mGluRI, which modulated K2P channels through a G-protein-triggered mechanism (Figs. 5 and 6). Our results point to a mechanism involving TASK and TREK K2P channels, but other channels could play a role as well since our pharmacological screen could not exhaustively test specific interactions.

Although this mechanism may seem surprising, it does align with other observations. First, synaptic responses elicited by strong higher-order thalamocortical stimuli were shown to consistently contain mGluRI components[21,22]. Second, activation of mGluRs in neocortex and hippocampus is known to modulate various K+ channels[81–86], although Ca2+ and other channels can also be affected[87]. Third, signaling pathways from mGluRs to K2P channels have been identified using reduced expression systems[88,89] and in cerebellum granule cells[46,47,90].

K2P channels are responsible for leak and background currents, which are largely voltage-independent[91,92]. As such, they play an important role in regulating the resting membrane potential and hence a neuron's excitability. By reducing the number of background potassium channels that are in an open state, mGluRI signaling could depolarize the membrane by a few mV. The metabotropic action of the receptors together with their extrasynaptic location might make this a delayed, slow, yet all-or-nothing effect. The delays and the kinetics of the DSDPs that we detected upon POm stimulation are consistent with such a mechanism. It is important to note that the interaction between mGluRI and K2P are intricate, and may involve antagonistic signaling cascades[46,47]. It will be interesting to investigate if the various interactions depend on the molecular composition and if they are selective for synaptic or neuronal types. Furthermore, other GPCR-signaling pathways may also converge onto the K2P channels, e.g. through cholinergic signaling[47].

Our patch-seq experiments suggest that the gene expression profiles of neurons with BT-like appearances differs from those of ST-like neurons, which supports the notion that they have distinct morphological, connectivity, and functional profiles. Interestingly, BT-like neurons have on average higher levels of mGluR transcripts as compared to ST-like neurons, which is supported by data from the Allen Brain Atlas (mouse brain ISH data) showing higher transcript levels in upper L2/3. Thus, BT neurons may have a molecular disposition to mGluRI signaling. Our observation that the mGluRI-mediated modulation of the K2P channels only occurred for POm-to-BT inputs, and not by neighboring long-range synaptic inputs, also suggests that selectivity is regulated at the synaptic level. A combination of post-synaptic scaffold proteins may selectively recruit mGluRs, or their precise localization could be regulated through trans-synaptic interactions similar to the recruitment of presynaptic mGluR7[93] – although such interactions have as yet not been described for mGluR1 or 5. Synapse selectivity could also emerge from differential expression and trafficking of K2P family members or the intermediate signaling molecules[92,94] – although it is not clear whether localization can also be dendritic domain-specific.

### The role of POm-mediated mGluRI signaling in regulating excitability

The POm evoked mechanisms could increase the excitability of pyramidal neurons in two ways. First, the dendritic DSDPs bring the resting membrane potential into a more depolarized state, which may increase the probability that coinciding EPSPs cross the spiking threshold. DSDPs may also serve as critical precursors to NMDA spike generation as we show in conditions where activity is enhanced (Supplementary Fig. 14). In this respect, the POm-driven DSDPs are comparable to the cortical up-states that were previously reported upon photostimulation of thalamocortical circuits[95]. Second, the modulation of K+ channels may facilitate the transformation of electrogenic events originating in individual dendrites into global dendritic activity (i.e., simultaneous depolarizations in many dendrites)[33,96], which may promote the generation of action potentials[97–100]. Indeed, dendritic K+ conductance has been shown to inhibit the initiation of local supra-linear events, prevent the backpropagation of action potentials into the dendrites, dampen excitatory synaptic events, and more generally, decouple the dendritic from the somatic compartment[99,101]. Interestingly, thalamic activation (POm) of mGluRI has been found to be necessary for the coupling between the dendrites and soma of L5 pyramidal neurons[102,103]. Together with our findings, these observations suggest that a POm-driven block of K2P-conductance increases the excitability of distal dendritic compartments and thereby increases the transfer of depolarizations caused by other distal inputs from the dendrites to the soma. Overall, this aligns with studies showing that the activation of the POm causes a general increase in cortical excitability of the barrel cortex[24,71], and that the activation of postsynaptic mGluR5 receptors induces persistent firing in the prefrontal cortex[104]. Also, in line with these findings, we found that the local injection of mGluRI agonists and antagonists in vivo bidirectionally affected movement-associated activity of L2 neurons. In particular, the movement-related prediction from L2 neurons' activity increased upon the presence of mGluRI agonists and decreased with antagonists (Fig. 7). This strongly suggests that L2 pyramidal neuron activity, which likely depends on active feedback loops among others from POm, is mediated by mGluRI-associated mechanisms; possibly through the modulation of K2P channel opening. Using in vivo patch clamp recordings, we confirmed that mGluRI signaling can push membrane properties towards higher excitability. The regulation of excitability and dendritic coupling of various pyramidal neurons might rely on this mechanism. Recent work by the Larkum laboratory has shown that anesthetics cause dendritic decoupling and propose that this could be the underlying mechanism for the loss of consciousness[103]. Interestingly, K2P channels have been found to be positively modulated by some anesthetics[105,106]. This implies that POm inputs to L5 and L2/3 may play an important role in

modulating levels of consciousness, which could even be a general feature of higher-order thalamocortical inputs to various cortical areas.

# Methods

## Mice

C57BL/6 J wild-type (Charles River, Janvier Labs or born in house) and PV-Cre mice (MGI:3590684 https://www.jax.org/strain/008069, RRID: IMSR JAX:008069), aged 8 to 12 weeks, were group housed with littermates on a normal 12-h light cycle with food and water available *ad libitum*. All procedures were conducted in accordance with the guidelines of the Federal Food Safety and Veterinary Office of Switzerland and in agreement with the veterinary office of the Canton of Geneva (license numbers GE12219B, GE/74/18 and GE253A). Both males and females were used.

## Virus injection for ex-vivo electrophysiology and multimodal PatchSeq

C57BL/6 J or Parvalbumin (PV)-Cre mice, 8–12 weeks old, were anesthetized with isoflurane mixed with oxygen (3–5% induction, 1–2% maintenance), placed in a stereotaxic apparatus, and prepared for injections with craniotomies over the target injection regions. Deep anesthesia was assessed by absence of foot pinch reaction. The skin overlying the skull was removed under local anesthesia using Carbostesin (AstraZeneca) or Lidocaine (Streuli). Mice were then head-fixed with ear-bars and a nose clamp on a stereotaxic apparatus (Stoelting). Eyes were protected from drying with artificial tears. The body temperature was monitored with a rectal probe and was maintained at -37 °C using a heating pad (FHC) during surgery. Bilateral craniotomies were performed using an air-pressurized driller and injections (100–200 nl per injection site) were performed using a pulled glass pipette (10–15 μm diameter tip) mounted on a Nanoject II small-volume injector (Drummond Scientific). Injections were performed at a speed of 23 nl/s, separated by 2–3 min intervals, in POm (2.2 mm posterior to bregma, 1.2 mm lateral and 3 mm below the bregma), VPM (1.85 mm posterior to bregma, 1.75 mm lateral and 3.5 mm below the bregma), S1 (1.5 mm posterior to bregma, 3.5 mm lateral and 0.4 mm below the pial surface), M1 (1.54 mm anterior to bregma, 1.75 mm lateral and 0.5 mm below the pial surface) and S2 (0.7 mm posterior to bregma, 4.2 mm lateral and 0.3 mm below the pial surface). Depending on the experiment, AAV2/5-EF1a.eGFP.WPRE.RBG (University of Pennsylvania Vector Core; Cat # 105547-AAV5 RRID: Addgene_105547), AAV2-CB7.CI.mCherry.WPRE.RGB (University of Pennsylvania Vector Core; Cat # 105544-AAV2 RRID: Addgene_105544), AAVrg-pmSyn1.EBFP-Cre (AAV pmSyn1-EBFP-Cre was a gift from Hongkui Zeng; Addgene 51507-AAVrg, RRID: Addgene_51507), pAAVrg-CAG-tdTomato (pAAV-CAG-tdTomato was a gift from Edward Boyden; Addgene 59462-AAVrg, RRID: Addgene_59462), AAV2-rAAV.EF1a-DIO-hChR2(E123T/T159C)-eYFP (pAAV-Ef1a-DIO hChR2(E123T/T159C)-EYFP was a gift from Karl Deisseroth; Addgene 35509-AAV2, RRID: Addgene_35509), pAAV-Syn-ChrimsonR-tdT (pAAV-Syn-ChrimsonR)-tdTomato was a gift from Edward Boyden; Addgene 59171-AAV5; RRID: Addgene_59171, AAV2-CaMKIIα-hChR2-eYFP (pAAV-CaMKIIa-hChR2(H134R)-EYFP was a gift from Karl Deisseroth; Addgene 26969-AAV2, RRID: Addgene_26969) were injected with regards to the different experiments. All injections were bilateral. The pipette was left in place for 3–10 min before removing it from the brain. Mice were given analgesics (carprofen 5 mg/kg; TW Medical, #PF-8507) after surgery and monitored daily to ensure full recovery. Animals were then put back in their home cage to recover from the surgery. A minimum period of three weeks was allowed for viral expression before the animals underwent additional experimental procedures.

## Acute brain slice preparation

For electrophysiological recordings, mice were anesthetized with a ketamine/xylazine (100 mg/kg, 10 mg/kg) cocktail and were perfused intracardially with ice-cold high sucrose saline solution consisting of the following (in mM): 2.8 KCl, 1.25 NaH$_2$PO$_4$, 25 NaHCO$_3$, 0.5 CaCl$_2$, 7 MgCl$_2$, 7 dextrose, 205 sucrose, 1.3 ascorbate, and 3 sodium pyruvate (bubbled with 95% O$_2$/5% CO$_2$ to maintain pH at -7.4). A vibrating tissue slicer (Leica VT S1000, Germany) was used to make 250-μm-thick sections from 0.58 to 1.46 mm posterior to the bregma position. For obtaining S1 acute slice, the brain was removed and mounted to the stage of the vibratome, and sections were made coronally. Slices were held for 30 min at 35 °C in a chamber filled with artificial cerebrospinal fluid (ACSF) consisting of the following (in mM): 125 NaCl, 2.5 KCl, 1.25 NaH$_2$PO$_4$, 25 NaHCO$_3$, 2 CaCl$_2$, 2 MgCl$_2$, 10 dextrose, and 3 sodium pyruvate (bubbled with 95%O$_2$/5% CO$_2$) and then at room temperature until the time of recording.

For single-cell multimodal PatchSeq experiments, acute brain slices were prepared as described previously[107] with some modifications. Briefly, C57 BL6 mice, 3–4 weeks post-surgery, were perfused transcardially with ice-cold sucrose cutting solution (95% O$_2$/5% CO$_2$) containing (in mM): 75 sucrose, 85 NaCl, 0.5 CaCl$_2$, 4 MgCl$_2$, 24 NaHCO$_3$, 2.5 KCl, 1.25 NaH$_2$PO$_4$, and 25 glucose. Post perfusion, brains were surgically dissected and coronal slices (300 μm thick) were prepared using a Leica VT 1200S vibratome. Slices were incubated at 35 °C for 20 min in ACSF composed of (in mM): 125 NaCl, 2.5 CaCl$_2$, 1 MgCl$_2$, 26 NaHCO$_3$, 2.5 KCl, 1.25 NaH$_2$PO$_4$, and 25 glucose before recording.

## Dendritic whole cell recording in L2/3 pyramidal cells

The intracellular solution contained the following (in mM): 120 K-gluconate, 16 KCl, 10 HEPES, 8 NaCl, 7 Phosphocreatine, 2 K, 0.3 Na-GTP, 4 Mg-ATP, pH 7.3 with KOH[108,109]. Biocytin (Vector Laboratories; 0.1–0.2%) was also included for histological processing and *post hoc* cell location determination. Considering these solutions applied, the equilibrium potential of potassium ions (K$^+$) was calculated using the Nernst equation at a physiological temperature of 36 degrees Celsius yielded an estimated reversal potential of approximately −105.8 mV. In some experiments, Alexa-594 (16 μM; Thermo Fisher Scientific, #A10428) was also included in the internal recording solution to determine the dendritic recording location relative to the soma as well as a first assessment of the cell morphology. Dendritic recordings were performed at approximately 157 ± 25 μm from the soma for BT neurons and 208 ± 43 μm for ST neurons. Data was acquired using a Multiclamp 700b amplifier and the Clampex11 (Molecular Devices) data acquisition software. Data were acquired at 10–50 kHz, filtered at 2–10 kHz, and digitized by an Axon Digidata 1550B interface (Molecular Devices). Pipette capacitance was automatically compensated for. Series resistance was monitored and compensated throughout each experiment and was 10–25 MΩ for somatic recordings and 15–40 MΩ for dendritic recordings. Recordings were discarded if series resistance increased by more than 30% during the recording. Voltages are not corrected for the liquid-junction potential (estimated as -8 mV). The acute slice was placed in a recorded chamber with a feedback temperature system set at 36 degrees and continuously perfused with oxygenated ACSF (flow rate 1.5 ml/min). For photostimulation, 1 or 5-ms long light pulses were delivered through the objective using the coolLED pE-300ultra (CoolLED Ltd.), delivering blue light (475 ± 23 nm) for activating ChR2 and/or amber light for activating ChrimsonR (575 ± 25 nm). The intensity of the LED was normally set to 1.6 mW/mm$^2$ (10% of the maximum LED power) for the blue light and 2.2 mW/mm$^2$ (30% of the maximum LED power) for the amber light unless stated otherwise in the text. Differential LED intensities were employed for Channelrhodopsin and Chrimson to mitigate potential off-target activation and minimize spectral overlap, thus enabling selective and robust activation of the targeted signaling pathways (Supplementary Fig. 5).

Depending on the experiments, the following drugs were perfused in the bath, together or sequentially as described in the main text and figure legends: APV (50 μM, Sigma Aldrich, A8054), bupivacaine (1 mM, Sigma Aldrich, PHR1128), QX-314 (1 mM, Tocris Bioscience, Cat.

No. 2313), Barium (1 mM, Sigma Aldrich, 217565), A1899 (100 nM, Tocris Bioscience, Cat. No. 6972), Fluoxetine hydrochloride (100 μM, Tocris Bioscience, Cat. No. 0927), IBMX (200 μM, Tocris Bioscience, Cat. No. 2845), LY367385 (50 μM, Tocris Bioscience, Cat. No. 1237), MCPG (500 μM, Tocris Bioscience, Cat. No. 0336), MPEP (10 μM, Tocris Bioscience, Cat. No. 1212), TTX (1 μM; Latoxan, L8502), 4AP (100 μM; Sigma Aldrich, A78403), NBQX (10 μM, Abcam, ab120046), bicuculline (10 μM, Tocris Bioscience, Cat. No. 0130), DHPG (10 μM, Tocris Bioscience, Cat. No. 0342), DL-threo-β-Benzyloxyaspartic acid (DL-TBOA, 10 μM, Tocris Bioscience, Cat. No. 1223). For some experiments, GDP-β-S (1 mM, Sigma Aldrich, G7637) was mixed with the intracellular solution. For all experiments involving dendritic recordings, the membrane potential was maintained at the resting membrane potential, except for the analysis of supralinear events (NMDA spikes), where dendrites were held at approximately −55 mV to facilitate event onset.

### Analysis of electrophysiological recordings
The parameters of the photostimulation evoked events, such as the rise time, amplitude, duration, were analyzed using built-in functions in Clampex (Molecular Devices).

To assess if a neuron exhibited NMDA spikes with the co-stimulation of a pair of inputs across 30 trials, we analyzed the distribution of the mean amplitude of the photostimulation evoked events and calculated a bimodality coefficient from the distribution as follows:

$$BC = (S^2 + 1)/\left(K + 3 \times \frac{\frac{(N-1)^2}{N-2}}{N-3}\right) \qquad (1)$$

where $N$ is the number of samples, $K$ and $S$ are the data kurtosis and skewness respectively, calculated using MATLAB functions (Mathworks). We considered that a distribution was bimodal when $BC$ was greater than 0.5. Subsequently, a $k$-means cluster analysis with $k = 2$ was applied to determine which trials displayed NMDA spikes, and this served to calculate the fraction of trials and the mean amplitude of NMDA spikes. In addition, NMDA spikes were calculated by multiplying the fraction of trials with NMDA spikes with their mean amplitude.

To detect DSDPs, post-stimulation events were automatically identified by detecting the changes in the membrane potential using a threshold that corresponded to 3 times the standard deviation of the baseline noise. This baseline noise was defined as the data below the median value of the whole trace. Traces were then low pass filtered at 100 ms and events were detected as above the threshold for at least 10 ms. The amplitude and duration of all the detected events were then extracted. We used a conservative threshold of 200 ms to separate short and long-lasting events; the latter being classified as DSDPs confirmed by visual inspection. We then calculated the DSDP probability as the likelihood of eliciting at least one DSDP within a post-stimulus period. The total DSDP duration corresponds to the average cumulative duration of DSDP per trial. The DSDP frequency was calculated as the number of DSDPs per second across all post-stimulus recordings.

### Sample collection for PatchSeq
Slices were continuously superfused with oxygenated ACSF maintained at 30 ± 0.3 °C via an in-line heating system. TdTomato positive L2/3 neurons in S1 were visualized using an upright microscope (BX51WIF, Olympus) equipped with a 40× water-immersion objective, infrared differential interference contrast (DIC) optics, and epifluorescence (GFP and mCherry filter sets, 470 and 565 nm LED sources). Digital imaging was performed using a CCD camera (SciCam Pro, Scientifica). Recording pipettes (resistance 2−4 MΩ) were pulled from borosilicate glass capillaries (1.5 mm OD, GC150TF-7.5, Harvard Instruments) using a Zeitz DMZ puller. Pipettes were filled with a RNase-free internal solution containing (in mM): 123 potassium gluconate, 12 KCl, 10 HEPES, 0.2 EGTA, 4 MgATP, 0.3 NaGTP, 10 sodium phosphocreatine, 20 μg/ml glycogen, and 0.4 U/μl recombinant RNase inhibitor (pH ~ 7.3). For marking the patched neurons for *post-hoc* identification, recording pipettes contained fluorescent dye Alexa Fluor 488 (10 μM). After establishing a GΩ seal, whole-cell configuration was achieved by rupturing the membrane with gentle negative pressure. Signals were amplified (Multiclamp 700B, Molecular Devices), digitized at 10 kHz, and analyzed using Igor Pro. Cells with a resting membrane potential below −60 mV and stable series resistance (Rs) less than 30 mOhm were selected. Several active and passive electrophysiological parameters were recorded before slowly aspirating the cytoplasmic content and nucleus of the patched neuron in patch pipette for further sequencing and transcriptomic analysis. Utmost care was taken to preserve the morphology of the neurons to enable post hoc identification and manual scoring as BT and ST neuronal subtypes (schematic attached). All the PatchSeq samples were flash frozen until library preparation and further sequencing using Smart-Seq v4, 3'DE kit at Genomic facility, UNIGE, Geneva.

### Morphological identification and reconstruction of recorded neurons
When approaching with the path pipette, L2/3 pyramidal neurons were discriminated by the orientation of their main apical dendrite visualized in bright field mode. A cell with an oblique apical dendrite would be selected as a putative BT neuron. On the contrary, a cell with a main apical dendrite going straight toward the pia would be classified as a putative ST neuron.

Once the electrophysiological recording was completed, the electrode was gently pulled back from the dendrite to avoid membrane ruptures and preserve the recorded neuron. Neurons were retrospectively reconstructed using Neurolucida (MBF Bioscience) to confirm their dendritic morphology (Fig. 1a, Supplementary Fig. 1a). To this end, each slice was transferred in paraformaldehyde (PFA) 4% in 0.1 M phosphate buffer saline (PBS) for 10−15 min, then stored in 0.1 M PBS at 4 °C up to 1 week until the beginning of the biocytin staining procedure. As previously reported[110], slices were washed in PBS, then incubated in 1% Triton for 30 min and in 0.5% $H_2O_2$ for 30 min. After PBS washing, slices were incubated with VECTASTAIN Elite ABC Horseradish Peroxidase kit (Vector Laboratories) for 48 h at 4 °C. Slices were washed again in PBS and reacted with the chromogen 3,3´-diaminobenzidine (DAB kit, Vector Laboratories). When the reaction was complete, slices were mounted with the Vectashield mounting medium (Vector Laboratories). Dendritic arborizations were reconstructed in bright field under a 100×/ 1.30 NA oil-immersion objective using a Neurolucida system (MicroBrightField). Only spiny neurons were included in the reconstructed pool. VIP spiny neurons[111] were discriminated based on their distinct electrophysiological passive and firing properties. Quantification of the length and number of branches was automatically extracted from the reconstructions. Basal and apical dendrites were defined automatically as all the dendrites that originate from below or above the centroid of the soma respectively (Supplementary Fig. 1).

### PatchSeq bioinformatic analysis
PatchSeq cells were selected for analysis based on the following quality control criteria: (1) Visual confirmation of viral labeling and neuronal health, (2) ≥10,000 total reads, (3) ≥25% reads mapped to exonic regions, and (4) <15% mitochondrial content.

Morphological cell types were assigned based on visual scoring of the fluorescently labeled neurons using the Alexa fluor 488 and tdTomato signals. Allen brain cell type database was used to build a reference transcriptomic database of L2/3 neurons from somatosensory cortex (SSC) of mouse. A dimensionality reduction was conducted on this reference dataset and integrated with in-house PatchSeq data using Seurat V3 to represent in UMAP 2D space (Supplementary Fig. 13).

## In vivo whole-cell recordings

Acute electrophysiological recordings were carried out on 6–35 weeks old male C57BL/6 J mice. In some mice, viral injections were performed following the procedure described above. Mice were first anesthetized by first using a mix of O2 and 4% isoflurane at 0.4 L.min⁻¹ followed by an intraperitoneal injection of MMF solution, consisting of 0.2 mg.kg⁻¹ medetomidine (Orion Pharma), 5 mg.kg⁻¹ midazolam (Sintetica), and 0.05 mg.kg⁻¹ fentanyl (Mepha) diluted in sterile 0.9% NaCl. Mouse body temperature was maintained at 37 °C through a feedback-controlled heating pad (FHC). Eye ointment was applied to prevent dehydration. Analgesia was provided by local application of 1% lidocaine (Streuli). Betadine and 70% ethanol were applied at the skin of mouse head for disinfection. After exposing the skull, a small metal plate (~2 cm × 3 cm) was attached at the right mouse barrel cortex (1.4 mm caudal from bregma, 3.5 mm lateral from midline; S1) using cyanoacrylate glue and dental cement. C1 or C2 barrel fields were then located using intrinsic optical imaging. For this, the skull was thinned using a pneumatic dental drill and the Ringer's solution (145 mM NaCl, 5.4 mM KCl, 10 mM HEPES, 1 mM MgCl2, and 1.8 mM CaCl2) was applied. Intrinsic optical imaging was performed through the thinned skull upon stimulation of C1 or C2 whisker to identify the corresponding barrel fields at S1[23,65]. Using a stable 100-W halogen light source, red light (through a 700-nm interference filter with a bandwidth of 20 nm) was illuminated for collecting tissue reflectance signal and green light (through a 546-nm interference filter) was illuminated for imaging the vascular pattern at the cortical surface. Images were acquired using Imager 3001 F (Optical Imaging), equipped with a fast readout, and low read-noise charge-coupled device (CCD) camera. The size of the imaged area was adjusted by using a combination of two lenses with different focal lengths (upper lens: 135 mm, f2.0, Nikon; bottom lens: 50 mm, f1.2, Nikon). The CCD camera focused onto a plane ~300 μm below the skull surface. Images were acquired at 10 Hz with a spatial resolution of 996 × 996 pixels in an area of ~3.5 × 3.5 mm. A glass capillary, which was attached to a piezoelectric bender actuator (PL-140.11, controlled by an E-650 driver, Physik Instrumente) triggered by a pulse stimulator (Master-8, A.M.P.I.), was placed to surround the C1 or C2 whisker and ~4 mm away from the skin for deflecting the whisker back-and-forth. Each imaging trial of 50 frames lasted 5 s, consisting of a 1-s baseline period, followed by a 1-s stimulus period (upon 20 whisker stimulations at 8 Hz for 1 s) and a post-stimulus period. Inter-trial intervals lasted 20 s to avoid contamination of current intrinsic optical signal by prior stimulation. Responses were visualized by dividing the stimulus signal by the baseline signal. The C1 or C2 barrel field at S1 could be identified by overlapping the intrinsic signal with the vasculature image using ImageJ software.

After intrinsic optical imaging, a small circular craniotomy of ~1 mm diameter was made using the dental drill over the C1 or C2 barrel field at S1 and the dura mater was removed. Agarose (0.5–2% in Ringer's solution) and a cover glass were applied on top of the craniotomy to dampen the tissue movement. Whole-cell patch-clamp recordings of L2/3 pyramidal neurons were obtained using two-photon laser scanning microscopy (2PLSM)-guided shadow-patching[112]. A 2PLSM, like the one described in 'Two-photon laser scanning microscopy' but equipped with galvanometric scan mirrors (Cambridge Technology) was used for visualizing the unlabeled cell somata ('shadows') and the patch pipette filled with internal solution containing a fluorescent dye.

Patch pipettes of 5–8 MΩ tip resistance were fabricated from borosilicate glass by using a vertical microelectrode puller (Narishige). The internal solution contained 135 mM K-gluconate, 4 mM KCl, 10 mM HEPES, 10 mM Na2-phosphocreatine, 4 mM Mg-ATP, 0.3 mM Na-GTP, 0.2% biocytin, and 25–50 μM AlexaFluor-488 or AlexaFluor-594 for pipette visualization. An excitation wavelength of 800 nm was used for visualizing AlexaFluor-488 and 920 nm or 980 nm for AlexaFluor-594.

The external solution, which was applied on top of the craniotomy, contained the Ringer's solution.

After break-in to establish a whole-cell configuration, intrinsic electrophysiological properties and membrane potential (Vm) were monitored in current-clamp mode. The initial access resistance, which was compensated using the bridge balance, was $13.57 \pm 0.17$ MΩ (mean ± s.e.m., $n = 18$ cells). Data were obtained using a Multiclamp 700B amplifier and a Digidata 1440 A digitizer, which were controlled by pCLAMP 10 software (Molecular Devices). Electrophysiological data were filtered at 6–10 kHz and sampled at 20 kHz. Vm recordings were not corrected for liquid junction potential.

Whisker stimulation was applied using a similar configuration of glass capillary, piezoelectric ceramic bender, and a controller, as that for intrinsic optical imaging, but it was triggered by Digidata 1440 A. C1 or C2 whisker was deflected back and forth for 100 ms by applying 2 V or 4 V onto the piezoelectric ceramic, which corresponded to ~0.6 mm or ~1.2 mm whisker deflection respectively, with 8-ms for ramp-up or ramp-down. Whisker stimulation was repeated every 8 or 10 s, and a 100-ms step current-pulse of −20 pA was applied in each trial to access the input resistance.

To test the effect of mGluRI on cell excitability and whisker-evoked responses, Vm responses upon whisker stimulation with only Ringer's solution in the bath were first measured $19.6 \pm 4.1$ min (mean ± s.e.m., range: 4.5–68.6 min, $n = 18$ cells) after break-in. MCPG (0.5 or 1 mM, Tocris), DHPG (5 mM, Tocris) or Ringer's solution was then topically applied in the bath, and after $16.0 \pm 4.3$ min (mean ± s.e.m., range: 5.2–59.0 min, $n = 13$ cells in MCPG and DHPG conditions) or $4.3 \pm 0.8$ min (mean ± s.e.m., range: 2.1–6.3 min, $n = 5$ cells in Ringer's solution condition) measurements of Vm responses upon whisker stimulation were repeated.

## Analysis of membrane excitability and whisker-evoked responses

Baseline Vm, threshold of up/down-state, and input resistance were calculated based on individual trials of whisker-evoked responses (Supplementary Fig. 17). Whisker-evoked postsynaptic potentials (PSPs) were of larger amplitudes when the baseline Vm before whisker stimulation was at a down-state compared to an up-state. A baseline Vm was calculated for each trial as the mean Vm in −5–0 ms before whisker stimulation. A threshold of up/down-state Vm fluctuations for each trial was defined as the 10th percentile of Vm plus 3 mV. Trials of baseline Vm at down-states ('down-state trials') were selected when the baseline Vm was below the threshold of up/down-state and more hyperpolarized than −50 or −40 mV. To access the input resistance of down-state trials, another baseline Vm was first calculated as the mean Vm in −5–0 ms before the −20-pA current injection. Input resistance was then accessed as the quotient of dividing the voltage between minimum Vm during −20-pA current injection and this baseline Vm by the current injection amount of −20-pA. Another threshold of up/down-state was defined as the 10th percentile of Vm plus 1.5 or 3 mV was applied for selecting down-state trials before current injection.

## Surgery for in vivo calcium imaging

Stereotaxic injections of adeno-associated viral (AAV) vectors were carried out on 6 weeks old male C57BL/6 J mice. Anesthesia was first induced by a mix of O2 and 4% isoflurane at 0.4 L.min⁻¹ followed by an intraperitoneal injection of MMF solution. A mix of AAV1-Flex-hSyn1-mRuby2-GSG-P2A-GCaMP6s-WPRE-pA and AAV9-CaMKII-0.4.Cre-SV40 (Addgene, 68720-AAV1 and 105558-AAV9 respectively) with a ratio of 20:1 was delivered to L2/3 of the right S1 at the approximate location of the C2 barrel-related column (1.4 mm posterior to the bregma, 3.5 mm to the right, −0.3 mm below the pial surface). A 3-mm diameter cranial window, prepared with a silicone port[113], was implanted, as described previously[114]. Imaging was performed after at least 2 weeks of viral expression.

## In vivo drug injections

After recording the spontaneous activity of L2/3 neurons in baseline conditions, the mGluRI agonist DHPG (50 mM[115,116]; Tocris Bioscience) or antagonist MCPG (500 μM[117]; Tocris Bioscience) was injected through the silicone port of the cranial window using a glass pipette. A volume of 100 nl was slowly injected right below the pia. Mice were left to recover for 15 min before placing them back under the microscope for recording.

## Two-photon laser scanning microscopy

We used a custom built 2PLSM mounted onto a modular in vivo multiphoton microscopy system (https://www.janelia.org/open-science/mimms-10-2016) equipped with an 8-kHz resonant scanner and a 16× 0.8NA objective (Nikon, CFI75), and controlled with Scanimage 2016b[118] (http://www.scanimage.org). Fluorophores were excited using a Ti:Sapphire laser (Chameleon Ultra, Coherent) tuned to λ = 980 nm at an approximate power of 25 mW. Fluorescent signals were collected with GaAsP photomultiplier tubes (10770PB-40, Hamamatsu) separating mRuby2 and GCaMP6s signals with a dichroic mirror (565dcxr, Chroma) and emission filters (ET620/60 m and ET525/50 m, respectively, Chroma). Prior imaging, mice were handled and accustomed to being head restrained under the microscope for 10–15 min over 4–5 days. Two imaging depths were acquired quasi-simultaneously at approximately 10 Hz using a piezo z-scanner (P-725 PIFOC, Physik Instrumente) for moving the objective over the z-axis. The two planes were set with a size of 350 × 350 μm (512 × 256 pixels) and positioned at 100 and 300 μm below the pia (i.e., the upper and lower L2/3).

## Image processing

Images were processed using custom-written MATLAB scripts and ImageJ (http://rsbweb.nih.gov/ij/). Lateral motion corrections were performed using the reference mRuby2 signal, from the red channel. Rigid lateral movement vectors were calculated using the NoRMCorre MATLAB toolbox[119]. Residual bidirectional scanning artifact vectors were calculated using a highest-pixel-line signal correlation between the two scanning directions on the entire frame. All calculated lateral motion corrections were applied on both the mRuby2 and GCaMP6s channels. For an unbiased extraction of the GCaMP6s fluorescence signals from individual neurons, regions of interest (ROIs) were drawn manually for each session based on neuronal shape using the mRuby2 signal. The fluorescence time-course of each neuron and channels were measured as the average of all pixel values within the ROI. Local neuropil signal was measured for each ROI and channels as the average of pixel values within an automatically defined ring of 15 μm width, 2 μm away from the ROI and excluding overlapping regions with surrounding ROIs. Residual axial movement corrections were applied using the fluctuations in the mRuby2 signal of the measured ROIs. To perform this correction, signal traces were initially filtered using an exponential moving average filter with a window size of 500 ms. Then, the mRuby2 signal trace ($FR_{cell\ measured}$) was rescaled to the GCaMP6s ($FG_{cell\ measured}$) signal trace by normalizing the values using their 8th percentiles ($min\,R$ and $min\,G$ respectively) and their median values ($medR$ and $medG$ respectively) over a rolling window of 180 s as:

$$FR_{cell\ rescaled}(t) = ([FR_{cell\ measured}(t) - minR(t-90:t+90)] \\ /medR(t-90:t+90)) \times medG(t-90:t+90) \quad (2) \\ + minG(t-90:t+90)$$

We used the median value for normalization to consistently compare the mRuby2 signal with basal GCaMP6s signal. The GCaMP6s signal was then corrected as follow:

$$FG_{cell\ corrected}(t) = FG_{cell\ measured}(t)/FR_{cell\ rescaled}(t) \\ \times medG(t-90:t+90) \quad (3)$$

The same operations were performed on the neuropil signal to obtain the neuropil corrected vector ($FG_{neuropil\ corrected}$). The true GCaMP6s signal of a cell body was then estimated as[120]:

$$F(t) = FG_{cell\ corrected}(t) - r \times FG_{neuropil\ corrected}(t), \text{with } r = 0.7 \quad (4)$$

Normalized calcium traces $\Delta F/F_0$ were calculated as:

$$(F(t) - F_0)/F_0 \quad (5)$$

where $F_0$ is the 30th percentile of the whole $F$ trace. To determine if a neuron had changed its level of activity after drug injection, we calculated the effect size (or Cohen's d). The effect size corresponded to the difference in means between the before and after normalized calcium traces divided by the pooled standard deviation. To calculate the Cohen's d, we used a permutation test by shuffling 1000 times the datapoints between the two traces to create a null distribution of differences in means and standard deviations that would be expected under the null hypothesis of no difference between the two traces. We considered that a neuron changed its level of activity if the Cohen's d was more than 0.2, which corresponds to a small effect size.

## Anterograde labeling and analysis

AAV injections were performed following a similar procedure as for surgeries for in vivo imaging but without cranial window implantation. For these experiments, 20 nl of AAV1-mCaMKIIα-iCre-WPRE-hGHp(A) (UZH Viral Vector Facility, v206-1) was injected in either the VPM (1.8 mm posterior to bregma, 1.7 mm lateral and 3.5 mm below the bregma) or in the POm (2.2 mm posterior to bregma, 1.25 mm lateral and 3 mm below the bregma). A second injection of 200 nl of AAV2-hSyn-DIO-eGFP (pAAV-hSyn-DIO-EGFP was a gift from Bryan Roth, Addgene 105558-AAV9, RRID: Addgene_105558) at the approximate C2 barrel coordinates (1.4 mm posterior to bregma, 3.5 mm to the right and 0.3 mm below the pial surface). After 3–5 weeks of viral expression, mice were perfused, and brain slices were cut at a thickness of 300 μm. Images stacks of 650 × 650 × ~300 μm, with a voxel size of 0.3 × 0.3 × 1.5 μm, were acquired using 2-photon laser scanning microscopy (see above) tuned to 910 nm and a 25× 1.1NA objective (Nikon) at an approximate power of 50 mW. The positions of the soma were manually marked, and the pia position was automatically defined using a custom-written script in MATLAB.

## Decoding analysis

Snout movement recording was performed under 930 nm infrared illumination (M940L3, Thorlabs) using a 20 Hz infrared sensitive camera and the FlyCap acquisition software (FLIR Systems). Snout position was extracted from the raw video frames using the DeepLabCut v2.2.0.2 tracking algorithm[62]. In brief, a model was created using hand-annotated sample frames of the position of the snout in different imaging sessions. The model was then applied to determine the position of the snout in each frame of all the videos. The movement was calculated as the sum of the absolute derivatives of the x and y positions coordinates and applied a low-pass filter at 1 Hz.

A random forests machine-learning algorithm was used to decode the snout movements from the activity of single neurons. Given the slow kinetics of calcium transients captured by the GCaMP6s sensor, spiking rates were inferred from the $\Delta F/F_0$ trace and used as input to the algorithm, which allowed to temporally match fast motor movements to neuronal activity. For this we used a fast nonnegative deconvolution method (https://github.com/jovo/oopsi)[121] with

variable background fluorescence estimation and a $K_D$ of 144 nM[121]. For the algorithm to capture differences in activity levels between neurons, the activity traces from the movies before and after drug injections and of all neurons recorded were concatenated before inferring spikes. Both neuronal activity and behavior traces were resampled at 20 Hz. To account for putatively preceding pre-motor and/or following sensory-related activity in S1 relative to behavioral events, the neuronal activity traces were shifted negatively aasnd positively in time with a maximum shift of 250 ms. Thus, eleven time-shifted inferred firing rate traces (discretized in time bins of 50 ms) centered on zero time shift were used to predict instantaneous behavioral features and composed a vector $X_i$:

$$X_i(t) = [x_i(t - 250ms), \dots, x_i(t), \dots, x_i(t + 250ms)] \quad (6)$$

where $x_i(t)$ represents the inferred firing rates of the $i^{th}$ neuron at zero time shift. The ranger function of the ranger R package version 0.10.1 was used to construct regression forests, with the snout movement as the dependent variable and the binned inferred firing rates of a given neuron as predictors. Most arguments of the function were kept at default settings, except the following: the number of trees was set to 128, the minimum size of terminal nodes was set to 2, the number of predictor variables randomly sampled at each node split was set to the maximum between 1 or the third of the number of predictors, and the variable importance mode was set to "impurity". To obtain a prediction for all trials, 5-fold cross-validation was applied by training the algorithm on 80% of the data (i.e., training set) and evaluating it on the remaining 20% of the data (i.e., test set). For each neuron, the decoding accuracy was assessed by computing the Pearson's product-moment correlation coefficient between the observed and predicted behavioral event fluctuations. To verify that the measured change of PP value was not influenced by variations in noise level in the calcium activity traces, the noise power before and after drug application was calculated. For this, calcium activity signals were mean centered and transformed into the frequency domain using the Fast Fourier Transform (FFT), to compute the Power Spectral Density (PSD). A noise band between 0.5 and 5 Hz was defined to represent higher frequency noise. The FFT values within this noise band were isolated, and an inverse FFT was performed to obtain the noise trace in the time domain. Finally, noise power was calculated as the sum of the squared magnitudes of these noise components.

## Statistical analysis

All data are expressed as the mean ± s.e.m. unless stated otherwise. No data sets were excluded from analysis. For data obtained from electophysiological recordings, before applying the Student $t$-test, a QQ plot was generated, and the Shapiro-Wilk test was performed for each pool of data to confirm a normal distribution. Wilcoxon rank-sum test was used to compare two independent samples for non-normal distributions; Wilcoxon signed-rank test was used to conduct a paired difference test of repeated measurements on a single sample to assess whether their population mean ranks differ. Statistical analyses were performed using Prism (GraphPad), Origin (OriginLab), or MATLAB. A result was considered significant if $P < 0.05$. Power analyses were conducted using G*power and reported as Type II error probability β.

## Reporting summary

Further information on research design is available in the Nature Portfolio Reporting Summary linked to this article.

## Data availability

The data used to generate the figures is freely available at the CERN data repository Zenodo https://zenodo.org/communities/holtmaat-lab-data/ with https://doi.org/10.5281/zenodo.10210325. Source data are provided with this paper.

## Code availability

The principal Matlab code that was used for data analysis is freely available at the CERN data repository Zenodo https://zenodo.org/communities/holtmaat-lab-data/ with https://doi.org/10.5281/zenodo.10210325.

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

## Acknowledgements
We would like to express our gratitude to Dr. Fodoulian for generously providing the random forests code, which significantly enriched the computational aspects of our manuscript. We would like to thank Julien Prados and the bioinformatics platform for their help in the analysis of the PatchSeq experiments. We also want to extend the appreciation to the entire Holtmaat group for their valuable comments and constructive feedback, which have been instrumental in shaping and refining the development of this project. We would like to thank Elodie Husi, Sébastien Pellat and Raphaël Thurnherr for their technical support. This work was supported by the Swiss National Science Foundation Grant Numbers 31003A_173125 and 310030_204562 and the International Foundation for Research in Paraplegia (chair Alain Rossier to AH).

## Author contributions
Conceptualization, F.B., R.C. and A.H.; Methodology, F.B., R.C., F.M., S.P. and A.H.; Investigation, F.B., R.C., C.M., I.C., D.V.O. and N.M.; Formal Analysis, F.B., R.C., I.C., D.V.O. and T.B.; Resources, A.H.; Writing – Original Draft, F.B., R.C. and A.H.; Writing – Review & Editing, F.B., R.C. and A.H.; Visualization, F.B., R.C. and A.H.; Funding Acquisition, A.H.; Supervision, A.H.

## Competing interests
The authors declare no competing interests.
