## [Transparent Peer Review file · Nature Communications]

Thalamocortical feedback selectively controls pyramidal neuron excitability.

Corresponding Author: Professor Anthony Holtmaat

Version 0:

Reviewer comments:

Reviewer #1

(Remarks to the Author)

In the manuscript entitled Thalamocortical feedback selectively controls pyramidal neuron excitability, the authors carry out a series of experiments investigating the functional consequences of distinct synaptic afferent pathways on the excitability of two populations of layer II/III pyramidal neurons somatosensory cortex. The authors have used a multi-prong approach to address these important issues regarding synaptic integration involving distinct afferent pathways. The combined use of optogenetics, dendritic electrophysiology, and pharmacology, in conjunction with both *ex vivo* and *in vivo* approaches is extremely powerful. The results clearly show distinction between broad-tufted (BT) and slender tufted (ST) layer II/III pyramidal neurons with respect to their responsiveness to afferent synaptic inputs. The present organization of the manuscript is sometimes a bit difficult to follow the results, because nearly all of the quantified data is buried within each figure legend of primary and supplemental figures. The figures are well laid out and clear regarding the results; however, some of the "representative data" stray from the median/mean. While overall this is an important and significant series of experiments, there are a few issues that could be clarified or addressed.

1. For investigating the relative strength of different pathways on a single neuron (Figure 2), two different opsins were used. From the methods, only a single intensity was used for each opsin. In addition, the LED intensity for each of the opsins was also different. It would be helpful to understand why the given intensity was used, why are they different for each opsin? Was there an intensity series performed on the cells to delineate maximum response?

2. In figure 3, the authors nicely show that co-stimulation of two different pathways can produce differential excitation of an NMDA "spike" in the apical dendrites of BT versus ST neurons. What is unclear is whether a monosynaptic pathway with increasing stimulus intensity could produce an NMDA spike. For example, does increasing intensity of POM stimulation alone evoke an NMDA spike in a BT neuron? Similarly, does increasing VPM stimulation intensity lead to an NMDA spike in ST neurons? The converging hypothesis is interesting but is this simply a threshold issue in the dendrites, or requires summation of distinct pathways which would lead to two different interpretation of function.

3. The presentation and discussion of the "plateau potentials" are somewhat difficult to follow.

- In multiple figures (4e,i; 5d,e,g; 6b,e), the authors refer to "plateau frequency (Hz)", but it is unclear what this frequency is referring to. Does the tetanic stimulation (5 pulses) lead to repetitive occurrence of the plateau potentials? If so, that is not clear in any of the figures.
- Is there a functional or statistical reason for choosing 200 ms duration for distinction between plateaus and "short events"?
- The reversal of plateaus in Fig. 5a is not very clear. The plateaus in this example are 1 mV at best, but mean data (Fig. 5b) looks much more robust. Is there a better example to illustrate this with? Similarly, these plateaus look very different than all others illustrated and also lack the fast, short lasting responses to the optogenetic stimulation (a similar issue is the responses in Fig 6d).
- The functional significance of these plateaus is difficult to understand considering they are not consistently evoked, and the latency to response and duration, vary greatly.
- While the *in vivo* work clearly show a nice mGluR-dependent increase in sensitivity and enhanced output of the later II/III

neurons, it is unclear if this is related to plateau potentials, presumably in the dendrites, or a general mGluR-dependent depolarization of the pyramidal cells. Can the authors provide further evidence indicating the functional aspect of the 1-5 mV plateaus?

Reviewer #2

(Remarks to the Author)

The manuscript by Brandalise, Chéreau et al. utilizes ex vivo dendritic electrophysiology and optogenetic stimulation as well as in vivo calcium imaging and pharmacology to evaluate the role of different afferent inputs to different types of L2/3 pyramidal neurons in mouse somatosensory cortex. The authors use somato-dendritic anatomy based on biocytin reconstructions to provide evidence for 2 classes of L2/3 neuron: BT and ST, which correspond to previous observations in the field. They show that these neurons differ in the amount of axo-dendritic overlap with VPM, PoM, M1, and S1 inputs. This motivates their subsequent ex vivo patch clamp experiments to characterize the relative size and impact of these different inputs on BT vs ST neurons. They report preferences in targeting, and also observed an intriguing and novel slow potential only in BT neurons in response repeated PoM input. Pharmacology is used to identify the underlying K2P channels, mGluR1 and the G-protein pathway mediating these fascinating delayed potentials. The authors propose that the activation of mGluR1 decreases K⁺ conductance by modulating K2P channels, thus increasing dendritic excitability and reducing input attenuation towards the soma, augmenting the responses of other inputs. The authors try to connect this ex vivo mechanism to in vivo circuit function using mGluR1 pharmacology during 2P imaging in whisking mice. While the impact of the BT-PoM-mGluR1 mechanism on somatosensation and associated circuit processing is not clear, this work presents exciting new data and ideas for pathway-preferential modulation of different streams of input in L2/3 of cortex. Experiments are generally well-conducted, and the findings are novel, intriguing, and of broad interest. Some improvements should be made to strengthen the authors' claims and to potentially enhance impact, as detailed below.

Specific points:

1. The separation of BT and ST cells depends exclusively on biocytin-based dendritic anatomy. While these data look very compelling, this approach is quite old-fashioned now with the advent of powerful (and relatively easy) genetic approaches. The manuscript would be strengthened by evaluating the underlying genetic (e.g. transcriptomic) profile of BT and ST cells via something like patch-seq, at least via some data mining of the Allen Institute's V1 resources or other open genetics datasets. It would be very useful for the field to know what these BT and ST cells correspond to molecularly. The authors should evaluate and discuss how available/described Cre lines do or do not overlap with these two cell types.
2. It is remarkable that the authors use dendritic recordings for many experiments, but do not make any points about comparing dendritic and somatic properties between the two neuron types (AP shape, sag, Ri, etc). This is quite confusing, particularly in ED Fig. 1d & e. Readers will assume this is from somatic recording; there is no information in the figure or legend about recording location. These values will likely all be dramatically affected by recording location. AP thresholds look very high in the example traces in d – presumably because these examples are from quite far out in the dendrite? The authors should be sure to be clear about this to prevent confusion and misinterpretation.
3. Along these same lines, a panel in Fig 2 showing the distributions of recording locations along the somato-dendritic axes of the two cell types would be very useful in making it clear to readers that the following data comes from dendritic recording. I missed this on my first brief scan through the manuscript. The authors should certainly emphasize it, both for clarity reasons and to show off: dendritic recording from adult mouse L2/3 is almost unheard of! This is a heroic experiment: the authors should ensure that readers recognize and appreciate the over the top level of effort and rigor here!
4. The title of Fig. 2 could be potentially misleading – “single afferent” may be interpreted as the authors claiming they are activating a single axon, which they are clearly not. They should consider rewording.
5. The polysynaptic contamination in Fig. 2 makes these data hard to interpret. The main point the authors are trying to make about the different inputs and how strong they are at the different cell types requires monosynaptic specificity. The polysynaptic results are interesting, but distract from this first and most important point. The authors should show the monosynaptic results (currently in ED Fig. 3) first in this Figure 2, then show the polysynaptic results. More monosynaptic experiments are likely necessary, as the sample numbers in ED Fig. 3 are quite small.
6. Figure 3c shows the percentage of cells displaying at least one NMDA spike when stimulating pairs of presynaptic afferents together, but the underlying distribution is not shown. How many stimulation pairings were performed? How many of these stimulations failed to elicit NMDA spikes?
7. A minor semantic quibble: the authors state, based on Fig. 3, that “Our data show that these NMDA spikes are L2/3 neuron type and input selective.” These data do not show that: there are NMDA spikes in both cell types for all pairs of inputs. The data show biases or preferences according to cell type and input pair, not true selectivity.
8. I think the authors should change or at least alter the name they use to describe the slow depolarizing events mediated by K⁺ channel closure. “Plateau potential” is already highly overused in the literature for many different processes and is thus highly ambiguous/non-specific. The authors should not add to this confusion and should develop their own more defined term for their exciting new phenomena! Perhaps something like “delayed sustained dendritic potentials (DSDP)” or

something along those lines? This will help the authors should avoid confusion with prior usages of “plateau potential”, as well as the emerging colloquial usage now often associated with the depolarizing envelope underlying large somatic burst firing events in hippocampus and cortex in vivo.

9. I would encourage the authors to expand panel b in Fig. 4 – this is important and exciting data that is key for the point about the potentials, but it's so small it is hard to see.

10. It is unclear what holding potential(s) are used throughout the manuscript. While main text and ED Figure 5 states that the membrane potential was held at -55 mV for most of the experiments in this study, this should be clearly indicated in main text for each experiment, and in the figures themselves and respective legends. -55 mV is not what most readers will expect, so it is necessary to provide them with the appropriate information in each figure.

11. Line 266 states the potassium reversal potential was estimated to be around -100 mV, with the only explanation being “see methods”. The methods are missing such explanation.

12. The authors use a cocktail of bupivacaine + QX-314 + barium that they state is a “broad-spectrum cocktail of K2P channels blockers” (Fig 5c, d). However, this cocktail also blocks Na⁺ channels; the main text should be modified accordingly to prevent misleading the reader. In support of this cocktail, the authors cite Patel & Honoré (2001) which deals with K2P channel opening by volatile general anesthetics, but bupivacaine and lidocaine (QX-314) are neither general nor volatile anesthetics. It may be more appropriate to cite Lesage & Lazdunski (2000), which does indeed support TASK channel blockade by lidocaine and bupivacaine.

13. The example used in Fig. 5a is not particularly compelling – the delayed potentials are not very obvious at -60 mV. Here and at -125 mV they look like they could be noise or the recording falling apart, unlike in Fig. 4a (or 5c), where it is very convincing and robust. More samples are required in general for Fig. 5: each of the conditions is only n = 3 or 4. At least 5 seems necessary for appropriate statistical rigor. Numbers of mice should be reported too. Fig 6 has this issue too – more N required.

14. The in vivo experiments in Fig. 7 are not convincing due to the lack of specificity of the pharmacological manipulations. I applaud the authors' ambition and desire to connect their ex vivo experiments to in vivo circuit operation during behavior, but broadly activating or inhibiting mGluR1 in the cortex is going to alter many, many different (and potentially competing and/or interacting) processes. mGluR1 receptors are expressed at many other pre- and post-synaptic (and maybe glial or other non-neuronal cell) sites. While the results observed in Fig. 7 appear to mostly be consistent with what the authors predict for their BT-PoM slow potentials, the experiment is far too indirect and messy to provide robust support for this mechanism's role in vivo. Many other processes may well be producing this effect. Some kind of more specific experiment is necessary here. Some ideas: a cell-type specific (or at least preferential) knockout or knockdown of mGluR1 (or some other component of the mGluR1-to-TASK/TREK channels signalling pathway); in vivo patch clamp recording and manipulation of BT-PoM slow potentials during whisking and pharmacological treatment; a cell-type specific pharmacogenetic experiment like DART. I recognize these are not easy experiments – I'm sure the authors can think of something better (more controlled, more specific, and/or easier) than these, which I just came up with off the top of my head. But I think something along these lines is probably necessary to connect the ex vivo mechanism to its in vivo impact. A model would definitely not suffice.

15. The authors claim that in vivo calcium imaging confirms the presence of mGluR1-dependent modulation of inputs, but the MCPG/DHPG pharmacology reveals mixed effects. Of note, only 13% of upper L2/3 and <1% of lower L2/3 neurons decreased their activity upon application of antagonist MCPG. The authors should explain why the effect is so mixed, especially when their argument hinges on mGluR1 increasing dendritic excitability.

16. Related to point 2, Figure 7 is missing any measure of statistical significance of the deviation from baseline activity under MCPG and DHPG. Are the changes in activity statistically significant? While prediction power seems to be affected under the experimental conditions, the reduction of power under MCPG could also be explained by an increase in noise and not necessarily a decrease in activity. This could be easily addressed by showing how “noise” levels change before and after application of MCPG/DHPG, and whether such noise has any correlation with the PP (similar to Ext. Data Figure 11). Additionally, the authors do not explain why MCPG increases the overall activity in vivo when the opposite is expected based on their ex vivo pharmacology results.

17. The random forests decoder employed suggests increasing the gain of BT neurons through specific thalamocortical afferents increases the correlation between snout movements and activity. If the authors were to demonstrate a behavioral output difference (like an increase in discriminability between whisker stimulation frequencies or something of that sort), the impact of the novel BT-PoM depolarization mechanism would be stronger.

18. Were any structural reconstructions from the in vivo calcium imaging experiments performed to confirm that imaging at 100 μm and 300 μm is an accurate proxy for imaging almost exclusively BT and ST cells? Could this explain some of the mixed effects in the in vivo pharmacology?

19. Do the authors describe the epochs used for prediction of snout movements? Was the entire dataset used? Was the prediction power stable across the recording?

20. Line 298 appears to incorrectly reference Fig 6c.

Reviewer #3

(Remarks to the Author)

In this very interesting manuscript Brandalise et al. have dissected the effects of thalamocortical projections stimulation on layer 2/3 pyramidal neurons apical dendrites from the mouse barrel cortex. They have used a panoply of state-of-the-art techniques, as optogenetics and dendritic patch-clamp recordings ex vivo (slices), plus 2-photon laser scanning microscopy of calcium activity in vivo, combined with pharmacology. Their findings indicate that higher-order thalamocortical inputs are able to modify L2/3 neurons excitability in a cell-type and input-selective mode. NMDAR spikes and plateau potentials (caused by mGluRI modulation of K2P channels activity) are the main mechanisms associated with this increased dendritic excitability. The in vivo experiments are in nice agreement with their ex vivo results, as upper L2/3 neurons (mainly BT) bidirectional activity changes (produced by pharmacological manipulation of mGluRI signaling) correlate with the mice snout movements.

Overall, the conclusions of this paper are well supported by the data, the manuscript is well-written, nicely organized, the quality and quantity of work is considerable, and these findings should be of broad interest for the scientific community. I only have some minor points that should be addressed or clarified.

- K2P channels are involved in the regulation of resting membrane potential, as the authors mention. When the experimenters blocked these channels (figure 5), did they observed a depolarization of the membrane potential in their recordings, comparing with the control condition? To some extent, this blockade could account maybe as a kind of "sustained" plateau potential, and maybe an increase in spontaneous activity has been observed in those recordings.
- The authors use a specific mGlu1 receptor antagonist (LY367385) and a generic group I / group II mGluRs antagonist (MCPG). Did they try a specific mGlu5 receptor antagonist as MPEP or MTEP?
- Panels c and e from Extended Data Fig. 9 show input resistance changes and the average and SEM data are included in the figure legend. The numeric data could be included as well for input resistance plots in Fig. 5f and Fig. 6c,f. It seems clear that blocking K2P channels (Fig. 5f) will increase the dendritic input resistance at all times. However, it is not as clear for me if the blocking of mGluRIs (Fig. 6c) should be reflected in an input resistance decrease. Blocking the mGluRIs will prevent the K2P closing in those conditions that promote plateau potentials (i.e., P_{Om} stimulation). But during the input resistance protocol (exemplified in Extended Data Fig.1, panel d), that stimulation is not happening, as I understand. Moreover, in Fig. 6f, dialysis of GDP-beta-S did not significantly reduce the input resistance of the dendrite. P values for 6c and 6f are close to 0.05 in both cases, but the change of this parameter, if any, it would be probably very small.
- Extended Data Fig. 10 could be improved performing for the data in panel b the same quantification done in Fig. 4c. The scatter plot for the P_{Om} stim. initial condition (previous to TBOA application) could be added as well.
- There is some degree of dissimilarity between some sentences in Extended Data Fig. 11 legend and the main text. In the figure legend, it is said that for the DHPG dataset, "little to no correlation were observed for the upper and lower L2/3 neurons"; in the main text, "the PP of upper L2/3 neurons under control conditions moderately correlated with the magnitude of increase in activity under DHPG". The R value for upper L2/3 is 0.1 (DHPG), and R=-0.11 for lower L2/3 in MCPG. However, for lower L2/3 (MCPG), both main text and figure legend claim little to no correlation. Maybe the "moderately correlated" rating for upper L2/3 (DHPG) in the main text could be tempered.

Minor details (typos, etc.):

- Lines 108-109: "(Extended Data Fig. 1b,c)" → actually corresponds to "(Fig. 1c,d)"
- Line 196: "distribution of the evoked PSP amplitudes" → the parameter mentioned in figure 3 and Extended Data Fig. 6 as "event size" or "amplitude" corresponds to PSP peak area (mV.ms) rather than PSP peak amplitude (mV), as I understand from the figures and values. Maybe this could be clarified in the text.
- Line 264: "at -120 mV" → "at -125 mV" (measurements were done at -60, -100 and -125 mV)
- Line 298: "(Fig. 6c)" → actually corresponds to "(Fig. 6d)"
- Line 324: "(Extended Data Fig. 10A,B)" → "(Extended Data Fig. 10a,b)", lowercase for the panels letters
- Line 901: "500 μm, Tocris Bioscience" → "500 μM, Tocris Bioscience"
- Line 1175: "The bath application of 50 mM APV" → "The bath application of 50 μM APV"
- Line 1212: "S1 intracortical vs. VPM: P = 0.055" → This comparison/cumulative distribution plot is not shown in panel g, but panel j later, in line 1217.
- Lines 1312-1314: "The passive properties were assessed by analyzing the changes in current to a hyperpolarizing step from -70 mV to -80 mV (bottom)" → it should say "(right)" because it is placed at the right side of panel d.
- Page 61, Extended Data Fig. 9: in panel a, "10 μm" should be changed to "10 μM" (appears twice).

Version 1:

Reviewer comments:

Reviewer #1

(Remarks to the Author)

In the manuscript Thalamocortical feedback selectively controls pyramidal neuron excitability, Brandalise et.al. investigate the functional consequences of distinct synaptic afferent pathways on the excitability of two populations of layer II/III pyramidal neurons in the mouse somatosensory cortex. Having reviewed the previous submission of this article, the authors have done an excellent job of addressing all issues raised from the original version. Having completed multiple experiments as well as clarification of prior issues, this study provides key insight in the dendritic integration of layer 2/3 neurons. I have no further issues with the revised manuscript.

Reviewer #2

(Remarks to the Author)

The authors have addressed all of my concerns satisfactorily. I have no further comments other than well done. Its an intriguing and exciting set of findings, and I look forward to further work in this area.

Reviewer #3

(Remarks to the Author)

This latest version of the manuscript by Brandalise et al. has significantly improved the quality of the original article. In my opinion, all the comments, questions, and suggestions raised by the reviewers in the original version of the work have been clarified and appropriately answered. New data from new experiments have been provided to answer the reviewers' questions, and existing data have also been quantified and analyzed and are now shown in the new versions of the figures or as supplementary material. The quantity and quality of the new experiments is, in fact, exceptional and greatly appreciated for understanding the work. Therefore, I have no additional comments to make regarding this very interesting study.

We sincerely thank the reviewers for their thoughtful and constructive review of our manuscript. We greatly appreciate the fact that they all acknowledge the importance of our study (reviewer 1: "... this is an important and significant series of experiments..."; reviewer 2: "... findings are novel, intriguing, and of broad interest."; reviewer 3: "...the manuscript is well-written, nicely organized, the quality and quantity of work is considerable, and these findings should be of broad interest for the scientific community."). The reviewers' detailed feedback and suggestions for additional experiments have significantly strengthened the supporting evidence for the main conclusions of our study.

In response to the reviewer's, we have conducted most of the suggested experiments, and incorporated new datasets and analyses, which altogether reinforce the key findings of the manuscript ensuring a robust and comprehensive interpretation of the results. We believe these improvements have enhanced the manuscript's clarity, scientific value, and overall impact.

Reviewer #1 (Remarks to the Author):

In the manuscript entitled Thalamocortical feedback selectively controls pyramidal neuron excitability, the authors carry out a series of experiments investigating the functional consequences of distinct synaptic afferent pathways on the excitability of two populations of layer II/III pyramidal neurons somatosensory cortex. The authors have used a multi-prong approach to address these important issues regarding synaptic integration involving distinct afferent pathways. The combined use of optogenetics, dendritic electrophysiology, and pharmacology, in conjunction with both ex vivo and in vivo approaches is extremely powerful. The results clearly show distinction between broad-tufted (BT) and slender tufted (ST) layer II/III pyramidal neurons with respect to their responsiveness to afferent synaptic inputs. The present organization of the manuscript is sometimes a bit difficult to follow the results, because nearly all of the quantified data is buried within each figure legend of primary and supplemental figures. The figures are well laid out and clear regarding the results; however, some of the "representative data" stray from the median/mean. While overall this is an important and significant series of experiments, there are a few issues that could be clarified or addressed.

1. For investigating the relative strength of different pathways on a single neuron (Figure 2), two different opsins were used. From the methods, only a single intensity was used for each opsin. In addition, the LED intensity for each of the opsins was also different. It would be helpful to understand why the given intensity was used, why are they different for each opsin? Was there an intensity series performed on the cells to delineate maximum response?

We thank the reviewer for this important note. The use of different LED light intensities for the two different opsins are critical and were intentional. The intensities for each opsin were based on careful calibration experiments which are part of Extended Data Figure 5, which helped us to determine the optimal activation thresholds in our setup for

using both opsins simultaneously. To this end, we first measured the dose-response curves of cells expressing ChR2 or ChrimsonR, when applying blue (optimal for ChR2) and amber (optimal for ChrimsonR) LED illumination. As depicted in Extended Data Figure 5a-c, ChR2-expressing cells started to spike at high probabilities upon 5-10% of blue light, whereas ChrimsonR-expressing cells remain silent. Conversely, ChrimsonR-expressing cells started to spike at high probabilities upon 10-40% of amber light, which left ChR2-expressing cells silent. We characterized the levels for cross activation when recording postsynaptic potentials in pyramidal neurons upon photo-stimulation of excitatory inputs from ChR2-expressing POM afferents, and inhibitory inputs from ChrimsonR-expressing PV interneurons (Extended Data Figure 5d-g). This showed that 20% of blue light evoked excitatory but no inhibitory potentials, whereas 60% of amber light evoked inhibitory but no excitatory potentials. This reassured us that the LED intensities used in our experiments (10% blue, and 30% amber) yielded minimal to negligible off-target activation and cross-activation between the two opsins. To make this the reader a bit better aware of these important controls, we have now added the following sentence to the manuscript in the “Dendritic whole cell recording in L2/3 pyramidal cells” section of the Methods (line 960): “Differential LED intensities were employed for Channelrhodopsin and Chrimson to mitigate potential off-target activation and minimize spectral overlap, thus enabling selective and robust activation of the targeted signaling pathways”.

2. In figure 3, the authors nicely show that co-stimulation of two different pathways can produce differential excitation of an NMDA “spike” in the apical dendrites of BT versus ST neurons. What is unclear is whether a monosynaptic pathway with increasing stimulus intensity could produce an NMDA spike. For example, does increasing intensity of POM stimulation alone evoke an NMDA spike in a BT neuron? Similarly, does increasing VPM stimulation intensity lead to an NMDA spike in ST neurons? The converging hypothesis is interesting but is this simply a threshold issue in the dendrites, or requires summation of distinct pathways which would lead to two different interpretation of function.

These are interesting questions. In response to them we conducted additional experiments in order to test whether increasing photo-stimulus intensities of POM or VPM inputs alone can evoke an NMDA spikes in BT or ST neurons, respectively. We selectively expressed either a single opsin (ChR2 or ChrimsonR) in the POM or VPM and recorded evoked potentials in BT or ST neurons in response to moderate and high intensity stimulations. In both cases, we found that the PSP amplitudes increased with higher stimulation intensities, however the shape of the responses did not change. Even when supra-threshold depolarizations (evoking action potentials) were achieved in some trials, we could not detect any obvious NMDA spikes. In addition, we did not find a difference in the strength of the fast PSP (presumably predominantly AMPAR-driven) when comparing PSPs with and without NMDA spikes in BT neurons upon co-stimulation of POM and M1 afferents, nor in ST neurons upon co-stimulation of VPM and Intracortical S1 afferents, suggesting that NMDA spikes were not only produced upon large-amplitude PSPs. Together, these results suggest that the presence or absence of NMDA spikes under our experimental stimulus conditions was not simply a result of varying stimulus intensities or fast PSP amplitudes, but rather underscores the notion that NMDA spikes

occur in response to the activation of convergent pathways, hence synaptic cooperation. The results are now included in the Extended Data Figure 7.

Extended Data Fig. 7 | The induction of NMDA spikes does not depend on the stimulation strength and the initial PSP size (related to Fig. 3).

a,b Dendritic recordings in a BT (**a**) and ST (**b**) neuron following moderate and strong photo-stimulation of POM (**a**) and VPM (**b**) afferents. Whereas the amplitude of the PSP increases upon strong stimulation, the shape remains similar (see inset) and no NMDA spikes were induced, even in trials where action potentials were elicited. **c** Relationship between the fast and slow components of PSPs in 3 example BT neurons that exhibited NMDA spikes upon the co-stimulation of POM and M1. A *k*-means clustering analysis, with *k* = 2, was used to separate events with NMDA spikes (open circles) from regular PSPs (closed circles). The bar graphs show that the average strengths of the fast PSP components were similar between the 2 clusters, indicating that the initial PSP size does not influence the generation of NMDA spikes. **d** Same analysis for 3 example ST neurons that exhibited NMDA spikes upon the co-stimulation of VPM and intracortical S1.

3. The presentation and discussion of the “plateau potentials” are somewhat difficult to follow.

- In multiple figures (4e,i; 5d,e,g; 6b,e), the authors refer to “plateau frequency (Hz)”, but it is unclear what this frequency is referring to. Does the tetanic stimulation (5 pulses)

lead to repetitive occurrence of the plateau potentials? If so, that is not clear in any of the figures.

We apologize for the confusion. The tetanic stimulation variably led to the occurrence of plateau potentials – which we now term "delayed sustained dendritic potentials" (DSDP)", as per suggestion of Reviewer 2. They sometimes occurred multiple times in one trial but sometimes remained absent. To quantitatively capture their occurrence, we computed for each cell, the probability to detect at least one DSDP per trial, the total DSDP duration per trial, and the frequency over the total length of the post-stimulus traces over all trials. We had not been very clear about these quantifications. This has now been addressed more clearly in the Methods section under "Analysis of electrophysiological recordings" (line 1000) as well as in the main text (line 251).

- Is there a functional or statistical reason for choosing 200 ms duration for distinction between plateaus and "short events"?

We chose a 200-ms duration cutoff to confidently segregate long-lasting plateau potentials from AMPA or NMDA-mediated post-synaptic potentials. This cutoff was not arbitrarily chosen. AMPA-mediated post-synaptic potentials typically ranged from 20-50 ms whereas NMDA-mediated post-synaptic potentials were considerably longer yet rarely exceeded 100 ms in duration. In addition, similar types of sustained and long-lasting depolarizing events were shown to always be longer than 200 ms (Milojkovic et al., J Physiol 2007; Gao et al., J Neurophysiol 2021). Therefore, we considered that 200 ms was a conservative and safe value to distinguish between the plateaus and "short-lasting" events, and this allowed us to perform an unbiased automated detection analysis to compare the different conditions (such as in Figure 4).

- The reversal of plateaus in Fig. 5a is not very clear. The plateaus in this example are 1 mV at best, but mean data (Fig. 5b) looks much more robust. Is there a better example to illustrate this with? Similarly, these plateaus look very different than all others illustrated and also lack the fast, short lasting responses to the optogenetic stimulation (a similar issue is the responses in Fig 6d).

We agree. This was not a great example and Figure 5a was unfortunately based on a relatively small data set. Therefore, we decided to increase the number of recordings (n=6) from which we can present a more representative example. The example now clearly shows depolarizing potentials at -50 mV, which are comparable to the ones in other figures, as well as hyperpolarizing events at -120 mV. Collectively, the relationship between the holding potentials and DSDP amplitudes indicates a reversal potential of around -94 mV, which is consistent with our hypothesis that the plateaus are likely to be driven by a block of a K⁺ conductance. Regarding the example traces for which the photostimulation-evoked PSPs are barely visible, this was likely due to higher spontaneous activity, lower levels of opsin expression and a slightly more depolarized resting membrane potential that reduces the driving force of AMPA mediated responses.

- The functional significance of these plateaus is difficult to understand considering they are not consistently evoked, and the latency to response and duration, vary greatly.

Indeed, the plateaus are and remain somewhat enigmatic in many respects and as the reviewer notes, their onset latencies, duration and amplitudes vary greatly. There are various mechanisms that could underlie this.

First, we would like to note that the variability in plateau characteristics is not an uncommon finding in neuronal recordings and may reflect the intrinsically complex and dynamic nature of synaptic transmission and dendritic integration processes, as is exemplified by several previous studies: e.g., onset latency of dendritic plateaus can vary (Mease, R.A., Metz, M. and Groh, A., 2016. Cortical sensory responses are enhanced by the higher-order thalamus. *Cell reports*, 14(2), pp.208-215. **Fig 4**); e.g., the duration and magnitude vary depending on various factors such as the location of synaptic inputs (Oikonomou et al, 2014, Spiny neurons of amygdala, striatum, and cortex use dendritic plateau potentials to detect network UP states, *Frontiers in Cellular Neuroscience*, 8, 292, **Fig.6**); e.g., they can depend on dendritic morphology and network activity patterns (Waters and Helmchen, 2006, Background synaptic activity is sparse in neocortex, *Journal of Neuroscience*, 26, 8267, **Fig 3**).

Another possible explanation resides in the process and mechanism by which the mGluRs inhibit the K2P channels. First, the activation of mGluRs may depend strongly on extra-synaptic diffusion and therefore on insufficient clearing of glutamate, which will generate variability in the onset of the signaling cascade. Second, it has been reported that a fundamental step in the inhibition of K2P channels might be the accumulation of diacylglycerol (Wilke et al. 2014, Diacylglycerol mediates regulation of TASK potassium channels by Gq-coupled receptors, *Nature communications*, 5, 5540). The time constant of this process may strongly depend on other intracellular processes and molecular concentrations, as was shown in the aforementioned paper, generating variability in the time course of K2P channel closing. That said, we are aware that the plateaus are still enigmatic and that the explanations above leave quite some room for interpretation. We are following up on this work to further dissect and model this, but unfortunately those data are not yet available.

Nonetheless, to better guide the reader through the intricacies of this phenomenon and provide a better insight into the variability of these events, we have now included an extended data figure in which the onset latencies, durations and amplitudes are quantified (Extended Data Fig. 9 (see below)).

Extended Data Fig. 9 | Temporal Characteristics of DSDP Onsets and Durations Following POM Stimulation (related to Fig. 4).

a Probability distribution of the onset times of the first DSDP following POM stimulation (left histogram) and all DSDPs (right histogram) relative to the start of stimulation. The shaded blue area represents the period of POM stimulation. **b** Scatter plot of DSDP durations as a function of their onset times. Red circles represent the first DSDP, while black circles indicate secondary DSDPs. **c** On average, the duration of the first DSDPs were longer than the secondary DSDPs (1st DSDP : 659 ± 63 ms, $n = 52$ events; secondary DSDPs : 400 ± 38 ms, $n = 34$ events, means \pm s.e.m; $P = 0.0027$). **d** Same as **b**, but for the amplitude. **e** On average, the first DSDPs had larger amplitudes than secondary DSDPs (1st DSDP : 3.29 ± 0.55 mV, $n = 52$ events; secondary DSDPs : 1.40 ± 0.20 mV, $n = 34$ events, means \pm s.e.m; $P = 0.009$). **f** Scatter plot of first DSDP durations as a function of their onset times for individual cells, where each cell is represented in a different color. **g**

Same as **f**, but for the amplitude. **h** Coefficient of variation within cells for the onset, duration and amplitude of the first DSDP.

- While the *in vivo* work clearly show a nice mGluR-dependent increase in sensitivity and enhanced output of the later II/III neurons, it is unclear if this is related to plateau potentials, presumably in the dendrites, or a general mGluR-dependent depolarization of the pyramidal cells. Can the authors provide further evidence indicating the functional aspect of the 1-5 mV plateaus?

This is an important take-home point of our paper, which we would like to address here by elaborating on two key aspects.

1. Our *in vitro* electrophysiology data suggest that the mGluR-dependent DSDPs occur at dendrites, as we picked them up using dendritic recordings, which are best suited to detect distal dendritic depolarizations that otherwise would attenuate rapidly toward the soma (Magee et al., 2000; Waters et al., 2003; Larkum et al., 2007; Larkum et al., 2009). The relatively small and variable amplitudes of the recorded DSDPs (1–5 mV) likely reflect the distance between the event location and the recording site, but as such, and as alluded to by the reviewer, they may only have a limited direct impact on spiking. Nonetheless, we also showed that by prolonging the presence of ambient glutamate through the blockage of glutamate transporters using TBOA, we strongly favored the generation of NMDA-mediated spikes. These spikes were notably boosted within seconds after POM stimulation and were strongly associated with the DSDPs, i.e. they superimposed on them while they arose with delayed onsets after the stimuli (former Extended Data Fig. 10). Interestingly, TBOA did not cause this effect upon M1 stimulation which typically did not produce DSDPs. Together, this suggests that the DSPDs were the leading cause of the increased spiking as triggered by NMDAR-mediated events. These data are now presented in Extended Data Fig. 14.

2. Apart from the electrophysiological evidence for an association between DSDPs, NMDA spikes and APs, the recordings also strongly indicate that the underlying mechanism, i.e. the closing of K2P channels, leads to an increase in local input resistance (Fig. 5f; Fig. 6c; Extended Data Fig. 12c). We consider that this represents a very efficient mechanism for enhancing the gain of concurrent synaptic inputs that are coming from other afferents than the POM itself. We believe that it is this effect that underlies the modulation of neuronal activity as observed by our *in vivo* calcium imaging experiments of L2 and L3 neuronal populations. Thus, we interpret the spike modulation by mGluRI agonists and antagonists as an indirect readout of the changes in neuronal excitability, i.e. they decrease or increase the gain of incoming cortico-cortical signals, for example those from motor cortex – which are presumably strong during active whisking. However, as the referee correctly points out, the recorded calcium signals primarily reflect the output firing of these neurons, and it is therefore not very intuitive that this was linked to this modulating mechanism.

To strengthen the relationship between the *in vivo* and *in vitro* data, we have obtained and included an additional dataset in which we performed whole-cell patch-clamp recordings of cortical layer 2 pyramidal neurons in anesthetized mice. Consistent with our *in vitro* findings, these recordings show that mGluRI activation using DHPG

significantly increased the intrinsic excitability of L2/3 pyramidal neurons, as indicated by its depolarizing effect on the resting membrane potential and its elevating effect on the input resistance. Together, these findings, therefore, suggest that mGluRI signaling in a subset of upper L2/3 neurons enhances their propensity to generate spikes during whisking, which is most likely a result of their boosting effect on the intrinsic excitability. These new data are now included in the manuscript and can be found in Extended Data Fig. 17.

Extended Data Figure 17 | In vivo whole-cell recordings from superficial layer 2/3 pyramidal neurons reveal increased intrinsic excitability following mGluRI activation during whisker stimulation (related to Figure 7).

a Schematic of the experimental setup. Whole-cell somatic patch-clamp recordings were performed on superficial L2/3 PNs in S1 of anesthetized mice. Patched neurons were filled with Alexa Fluor-488 via the patch pipette to enable morphological characterization using 2PLSM. **b** Example of maximum intensity projections of the Alexa Fluor signal (top view, MIP x,y and side view, MIP x,z) from a patched neuron obtained from a 2PLSM z-stack. On the right side, MIP x,y view color-coded by imaging depth confirming the characteristic morphology of a BT neuron. Scale bars: 50 μ m. **c** Changes in baseline membrane potential following topical application of Ringer solution, DHPG (5 mM), or MCPG (0.5 or 1 mM). DHPG significantly depolarized the membrane potential ($n = 6$ neurons; $P = 0.05$, paired t -test), whereas Ringer's solution and MCPG had no significant effect (Ringer: $n = 5$ neurons, $P = 0.63$; MCPG: $n = 8$ neurons, $P = 0.22$). **d** Changes in the threshold of Up and Down states following drug application. DHPG significantly increased the threshold ($n = 6$ neurons, $P = 0.05$, paired t -test), whereas Ringer ($n = 5$, $P = 0.77$) and MCPG ($n = 8$, $P = 0.23$) had no significant effect. **e** Effect of Ringer, DHPG, and MCPG on input resistance. Ringer solution significantly increased input resistance ($n = 5$ neurons, $P = 0.03$, paired t -test), but individually, none of neurons exhibited a significant increase ($P > 0.05$, two-sample t -test). DHPG also significantly increased input resistance ($n = 6$ neurons, $P = 0.04$, paired t -test), while MCPG had no effect ($n = 8$ neurons, $P = 0.13$). For (**c–e**), circles represent individual trial measurements, with colors indicating different cells. Dashed lines denote the average for neurons that did not exhibit a significant change (two-sample t -test), while continuous lines indicate those neurons with significant changes. Averages of individual neurons were tested with paired t -tests and bars and error bars represent their mean \pm s.e.m. across cells. Overall, mGluRI activation with DHPG increased the excitability of BT neurons. However, blocking mGluRI signaling with MCPG did not reduce excitability, contrary to what might have been expected. This lack of effect may be due to a low basal level of mGluRI activation in control conditions. Indeed, as those recordings were performed under anesthesia, it is possible that the mGluRI signaling would be diminished and therefore could not be further reduced with MCPG. Furthermore, the effects of DHPG and the absence of MCPG effects align well with the conclusions drawn from the *in vivo* calcium imaging experiments (Fig. 7) where the overall activity of neurons did not significantly decrease.

We have made these two key points clearer in the discussion (lines 540 and 563)

Reviewer #2 (Remarks to the Author):

The manuscript by Brandalise, Chéreau et al. utilizes ex vivo dendritic electrophysiology and optogenetic stimulation as well as in vivo calcium imaging and pharmacology to evaluate the role of different afferent inputs to different types of L2/3 pyramidal neurons in mouse somatosensory cortex. The authors use somato-dendritic anatomy based on biocytin reconstructions to provide evidence for 2 classes of L2/3 neuron: BT and ST, which correspond to previous observations in the field. They show that these neurons differ in the amount of axo-dendritic overlap with VPM, PoM, M1, and S1 inputs. This motivates their subsequent ex vivo patch clamp experiments to characterize the relative size and impact of these different inputs on BT vs ST neurons. They report preferences in targeting, and also observed an intriguing and novel slow potential only in BT neurons in response repeated PoM input. Pharmacology is used to identify the underlying K2P channels, mGluR1 and the G-protein pathway mediating these fascinating delayed potentials. The authors propose that the activation of mGluR1 decreases K⁺ conductance by modulating K2P channels, thus increasing dendritic excitability and reducing input attenuation towards the soma, augmenting the responses of other inputs. The authors try to connect this ex vivo mechanism to in vivo circuit function using mGluR1 pharmacology during 2P imaging in whisking mice. While the impact of the BT-PoM-mGluR1 mechanism on somatosensation and associated circuit processing is not clear, this work presents exciting new data and ideas for pathway-preferential modulation of different streams of input in L2/3 of cortex. Experiments are generally well-conducted, and the findings are novel, intriguing, and of broad interest. Some improvements should be made to strengthen the authors' claims and to potentially enhance impact, as detailed below.

Specific points:

1. The separation of BT and ST cells depends exclusively on biocytin-based dendritic anatomy. While these data look very compelling, this approach is quite old-fashioned now with the advent of powerful (and relatively easy) genetic approaches. The manuscript would be strengthened by evaluating the underlying genetic (e.g. transcriptomic) profile of BT and ST cells via something like patch-seq, at least via some data mining of the Allen Institute's V1 resources or other open genetics datasets. It would be very useful for the field to know what these BT and ST cells correspond to molecularly. The authors should evaluate and discuss how available/described Cre lines do or do not overlap with these two cell types.

We thank the reviewer for this suggestion, which we have taken up. We have performed a PatchSeq experiment to map transcriptomic profiles of putative BT and ST neurons (33 cells total) onto the canonical transcriptomic subtypes provided by the Allen Brain Cell Types RNA-Seq database for the somatosensory cortex. We have added the results in a new figure (Extended Data Fig. 13). It is important to note that a thorough BT and ST classification (as in Figure 1) was not possible in this experiment since we could not fill the cells long enough for a detailed morphological analysis due to the technical constraints imposed by the PatchSeq protocol. Nonetheless, we were able to group the cells according to their BT or ST-like appearance, from which we were able to extract

various types of interesting information. First, a targeted analysis revealed a relative enrichment of transcripts of the metabotropic glutamate receptor genes in BT neurons as compared to ST neurons (Extended Data Fig. 13a,b). Interestingly, mGluR1 (Grm1) and mGluR5 (Grm5) gene transcript levels were higher on average in the BT group (Extended Data Fig. 13b), which aligns with our general finding that BT cells were more likely to display mGluR1-mediated modulation of excitability. The result is also congruent with the Allen Mouse Brain mRNA expression atlas, which shows that transcripts of mGluR1 genes are present in L2/3 neurons, with a bias to the upper segment of L2/3, which is enriched in BT cells (<https://mouse.brain-map.org/experiment/show/79591723>) (Extended Data Fig. 13c). Additionally, when we integrated the expression profiles of the two groups into the Allen Brain Cell Types RNA-Seq database (Yao et al. Cell 184, 3222), we observed that BT and ST neurons tend to separate towards distinct canonical transcriptomic L2/3 pyramidal cell subtypes (Extended Data Fig. 13d). In the Uniform Manifold Approximation and Projection (UMAP) map, the position of ST neurons was somewhat biased towards the Adamts2 class while BT neurons trended towards the Agmat class. Interestingly, this may be an indicator of functional differentiation and provide a handle for genetic access, as has been demonstrated by other studies (O'Toole et al. 2023, Neuron, 111, 2918; Condylis et al. 2022, Science 375, eabl5981). We would like to further explore in future work.

Extended Data Fig. 13 | Multimodal PatchSeq reveals differential molecular profiles as a function of morphological properties in mouse L2/3 neurons (related to Fig. 6).

a Heatmap displaying differential gene expression profiles between BT and ST neurons (expressed as row z-score), highlighting an enrichment of metabotropic glutamate receptor (mGluR) genes (Grm1, Grm2, Grm3, Grm5, Grm7, and Grm8) predominantly in BT neurons ($n = 15$) compared to ST neurons ($n = 18$). Rows correspond to individual mGluR genes, and columns represent neurons, color-coded by subtype. The clustering was generated using unsupervised random forest analysis, revealing distinct molecular signatures associated with BT and ST morphologies. **b** Average expression level, as \log_{10} -transformed reads per million (RPM), is significantly higher in BT neurons than in ST neurons for Grm1, 5, 3, 7 and 8 (Grm1: BT 0.72 ± 0.15 , ST 0.19 ± 0.07 , $P = 0.0027$; Grm5: BT 1.46 ± 0.11 , ST 0.82 ± 0.17 , $P = 0.0052$; Grm2: BT 0.10 ± 0.05 , ST 0.009 ± 0.009 , $P = 0.071$; Grm3: BT 0.65 ± 0.15 , ST 0.12 ± 0.05 , $P = 9.9 \times 10^{-4}$; Grm7: BT 1.45 ± 0.17 , ST 0.54

± 0.13 , $P = 1.6 \times 10^{-4}$; Grm8: BT 0.57 ± 0.13 , ST 0.20 ± 0.06 , $P = 0.013$, t-tests, $n = 15$ BT neurons, $n = 18$ ST neurons). **c** *In situ* hybridization images showing the expression of Grm1 and 5 in coronal sections of the mouse brain. Both Grm1 and 5 expresses at higher levels in the most superficial part of L2/3. **d** Uniform Manifold Approximation and Projection (UMAP) of canonical transcriptomic subtypes of cortical neurons from the somatosensory cortex (from Allen Brain Cell Types RNA-Seq database). PatchSeq data from 23 quality-controlled cells were integrated into this dataset. Classification into BT (bipolar tufted) and ST (simple tufted) subtypes was achieved through post-hoc manual scoring based on detailed morphological reconstructions of labeled cells. The transcriptomic subtypes identified included Agmat, Adamts2, Baz1a/Rrad, and Rorb clusters, as annotated on the UMAP plot. BT neurons (dark blue) show a tendency to map onto the Agmat cluster, whereas ST neurons (light blue) tend to be located within the Adamts2 cluster. This suggests that BT neurons share transcriptional similarities with Agmat-expressing excitatory subtypes whereas ST neurons align more closely with Adamts2-expressing subtypes. Each subtype is color-coded for clarity, and the location of BT (dark blue) and ST (light blue) neurons is highlighted within the UMAP to show their overlap with putative transcriptomic identities.

The corresponding Methods section was also included in the manuscript.

2. It is remarkable that the authors use dendritic recordings for many experiments, but do not make any points about comparing dendritic and somatic properties between the two neuron types (AP shape, sag, Ri, etc). This is quite confusing, particularly in ED Fig. 1d & e. Readers will assume this is from somatic recording; there is no information in the figure or legend about recording location. These values will likely all be dramatically affected by recording location. AP thresholds look very high in the example traces in d – presumably because these examples are from quite far out in the dendrite? The authors should be sure to be clear about this to prevent confusion and misinterpretation.

We thank the reviewer for this valuable feedback. We agree that is important to highlight that our data were obtained using dendritic recordings, which as the reviewer notes are quite critically different from somatic recordings. In fact, we believe that the dendritic recordings were essential for detecting distal dendritic depolarizations, such as the plateau potentials (now named ‘delayed sustained dendritic potentials’ or ‘DSDPs’ per this review’s suggestion [cf. point 8]) that otherwise would attenuate rapidly toward the soma (Magee et al., 2000; Waters et al., 2003; Larkum et al., 2007; Larkum et al., 2009). We have made a few efforts to observe DSDPs using somatic recordings, but this proved difficult for various reasons, including higher noise levels, smaller depolarizations, and interfering somatic inhibitory conductance. Therefore, we have not systematically pursued somatic recordings – ideally this would have to be done by dual recordings at dendrites and soma, which is even more complicated to achieve.

But better highlight that our data were derived from dendritic recordings, we have now included schematics in Extended Data Figure 1 and Figure 4, and explicitly state in the figure legends that the traces were obtained from dendritic locations. We also emphasize this aspect in the main text. As correctly noted by the reviewer, the AP thresholds in somatic recordings were typically lower than in the dendritic recordings. Indeed, the current injection required for eliciting APs in dendritic recordings is about the double of what is typically injected at the soma (somatic rheobase: 190 ± 49 pA, $n=5$ cells; dendritic rheobase: 410 ± 110 pA, $n=12$ cells). To avoid confusion among the readership, we now note in the figure legend that some of these properties as measured dendritically may be critically different from somatic recordings.

3. Along these same lines, a panel in Fig 2 showing the distributions of recording locations along the somato-dendritic axes of the two cell types would be very useful in making it clear to readers that the following data comes from dendritic recording. I missed this on my first brief scan through the manuscript. The authors should certainly emphasize it, both for clarity reasons and to show off: dendritic recording from adult mouse L2/3 is almost unheard of! This is a heroic experiment: the authors should ensure that readers recognize and appreciate the over the top level of effort and rigor here!

We thank the reviewer for this praise. We appreciate the recognition of the experimental challenges for conducting dendritic recordings from adult mouse L2/3 neurons. But indeed, the dendritic recordings allowed us to compare BT and ST cell responses at equidistance from the input sites in L1 in Figure 2, as indicated by similar rise times for these inputs between the two neuronal subtypes (Extended Data Fig. 2). As alluded to above under point 2, we have inserted at various places in the manuscript, the notion that the data are from dendritic recordings. In addition to the already mentioned schematics, we have also changed the title the e-phys methods section to “Dendritic whole cell recording in layer 2/3 pyramidal cells”.

4. The title of Fig. 2 could be potentially misleading – “single afferent” may be interpreted as the authors claiming they are activating a single axon, which they are clearly not. They should consider rewording.

We agree. We have updated the title to "Distinct input levels from different types of afferents on BT and ST neurons in S1", which more accurately reflects the experimental paradigm. We have also added in the legend that we apply “selective photostimulation of various populations of afferent inputs”.

5. The polysynaptic contamination in Fig. 2 makes these data hard to interpret. The main point the authors are trying to make about the different inputs and how strong they are at the different cell types requires monosynaptic specificity. The polysynaptic results are interesting, but distract from this first and most important point. The authors should show the monosynaptic results (currently in ED Fig. 3) first in this Figure 2, then show the polysynaptic results.

We agree that investigating the monosynaptic components would be interesting and informative. We have considered to perform extra experiments but respectfully decided against it. We feel that the provided comparison of monosynaptic inputs from VPM and POm between BT and ST cells nicely linked our results to earlier work in which these monosynaptic input maps were thoroughly characterized for cell types in different layers (e.g., Petreanu et al. 2009, *Nature* 457:1142; Sermet et al. 2019, *eLife* 8:e52665). We also think that a careful and complete mapping of various monosynaptic connections onto BT and ST cells would require a very large dataset and extra controls, which we feel is beyond the scope of the current study. The primary aim of our study is to highlight the difference between BT and ST cells in the total amount of integrated synaptic input that they receive from afferent pathways, as this will ultimately determine how these pathways interact when they are co-activated. By getting an insight into the level of combined mono- and polysynaptic EPSPs we can now seamlessly link these findings to the ability of the (combined) afferent inputs to produce NMDA spikes and DSDPs. Such an integrated approach would have been impossible by recordings in which monosynaptic inputs were isolated by TTX and 4AP, which greatly reduces the total input strength.

6. Figure 3c shows the percentage of cells displaying at least one NMDA spike when stimulating pairs of presynaptic afferents together, but the underlying distribution is not shown. How many stimulation pairings were performed? How many of these stimulations failed to elicit NMDA spikes?

We thank the reviewer for their comment. We specified in the Methods section that these recordings consisted of 30 trials of paired stimulations. In Figure 3c, the pie charts indicate the number of cells displaying at least one NMDA spike across all 30 trials. In addition, we show the probability of occurrence for individual cells (dots in the bar graphs) as well as their strength.

7. A minor semantic quibble: the authors state, based on Fig. 3, that “Our data show that these NMDA spikes are L2/3 neuron type and input selective.” These data do not show that: there are NMDA spikes in both cell types for all pairs of inputs. The data show biases or preferences according to cell type and input pair, not true selectivity.

We agree. The sentence has been rephrased (line 217) as follows: ‘Our data show that the occurrence of NMDA spikes in L2/3 pyramidal neurons is cell-type dependent and relies on the activation of specific combinations of inputs.’

8. I think the authors should change or at least alter the name they use to describe the slow depolarizing events mediated by K⁺ channel closure. “Plateau potential” is already highly overused in the literature for many different processes and is thus highly ambiguous/non-specific. The authors should not add to this confusion and should develop their own more defined term for their exciting new phenomena! Perhaps something like “delayed sustained dendritic potentials (DSDP)” or something along those lines? This will help the authors should avoid confusion with prior usages of “plateau potential”, as well as the emerging colloquial usage now often associated with

the depolarizing envelope underlying large somatic burst firing events in hippocampus and cortex in vivo.

We very much appreciate the reviewer's thoughts on this and their excellent suggestion. We agree that the term "plateau potential" could lead to confusion in the context of existing literature, including our previous work. Therefore, we have adopted the reviewer's proposed term, "delayed sustained dendritic potential (DPSP)," and have applied it consistently throughout the manuscript.

9. I would encourage the authors to expand panel b in Fig. 4 – this is important and exciting data that is key for the point about the potentials, but it's so small it is hard to see.

We agree with the reviewer. We have now expanded panel b in Fig. 4.

10. It is unclear what holding potential(s) are used throughout the manuscript. While main text and ED Figure 5 states that the membrane potential was held at -55 mV for most of the experiments in this study, this should be clearly indicated in main text for each experiment, and in the figures themselves and respective legends. -55 mV is not what most readers will expect, so it is necessary to provide them with the appropriate information in each figure.

We thank the reviewer for bringing up this important point. To clarify, for all experiments in the manuscript, dendritic recordings were held at the resting membrane potential, except for the experiments in which we induced supralinear events (NMDA spikes). In those specific cases, dendrites were held at approximately -55 mV to facilitate the onset of these events. We have now made sure that the holding membrane potentials are listed where necessary. We have also added the following sentence in the methods section (line 975): "For all experiments involving dendritic recordings, the membrane potential was maintained at the resting membrane potential, except for the analysis of supralinear events (NMDA spikes), where dendrites were held at approximately -55 mV to facilitate event onset."

11. Line 266 states the potassium reversal potential was estimated to be around -100 mV, with the only explanation being "see methods". The methods are missing such explanation.

We have addressed this issue by adding a detailed description in the methods section. Specifically, we applied the Nernst equation to calculate the equilibrium potential of potassium ions at a physiological temperature of 36 degrees Celsius. The result of this calculation yielded an estimated potassium reversal potential of approximately -105.8 mV.

12. The authors use a cocktail of bupivacaine + QX-314 + barium that they state is a "broad-spectrum cocktail of K2P channels blockers" (Fig 5c, d). However, this cocktail also blocks Na⁺ channels; the main text should be modified accordingly to prevent misleading the reader. In support of this cocktail, the authors cite Patel & Honoré (2001)

which deals with K2P channel opening by volatile general anesthetics, but bupivacaine and lidocaine (QX-314) are neither general nor volatile anesthetics. It may be more appropriate to cite Lesage & Lazdunski (2000), which does indeed support TASK channel blockade by lidocaine and bupivacaine.

Done.

13. The example used in Fig. 5a is not particularly compelling – the delayed potentials are not very obvious at -60 mV. Here and at -125 mV they look like they could be noise or the recording falling apart, unlike in Fig. 4a (or 5c), where it is very convincing and robust. More samples are required in general for Fig. 5: each of the conditions is only $n = 3$ or 4. At least 5 seems necessary for appropriate statistical rigor. Numbers of mice should be reported too. Fig 6 has this issue too – more N required.

We agree with the reviewer's observation. The sample size for the dataset in Fig. 5b has now been increased to improve statistical robustness. Additionally, more representative traces have been selected and are now displayed in Fig. 5a.

While we agree that increasing the sample size to $n=5$ or more for the rest of Fig. 5 and Fig. 6 might strengthen the statistics, we believe that the data provided with $n=4$ already demonstrate consistent trends across all conditions. The variability between samples is minimal, as evidenced by the tight clustering of data points. All the blocking conditions together provide a large n for the R_m , allowing robust statistics. We recognize the importance of reporting the number of animals used for each condition and we have now included this information in the revised figure legends.

14. The in vivo experiments in Fig. 7 are not convincing due to the lack of specificity of the pharmacological manipulations. I applaud the authors' ambition and desire to connect their ex vivo experiments to in vivo circuit operation during behavior, but broadly activating or inhibiting mGluRI in the cortex is going to alter many, many different (and potentially competing and/or interacting) processes. mGluRI receptors are expressed at many other pre- and post-synaptic (and maybe glial or other non-neuronal cell) sites. While the results observed in Fig. 7 appear to mostly be consistent with what the authors predict for their BT-PoM slow potentials, the experiment is far too indirect and messy to provide robust support for this mechanism's role in vivo. Many other processes may well be producing this effect. Some kind of more specific experiment is necessary here. Some ideas: a cell-type specific (or at least preferential) knockout or knockdown of mGluRI (or some other component of the mGluRI-to-TASK/TREK channels signalling pathway); in vivo patch clamp recording and manipulation of BT-PoM slow potentials during whisking and pharmacological treatment; a cell-type specific pharmacogenetic experiment like DART. I recognize these are not easy experiments – I'm sure the authors can think of something better (more controlled, more specific, and/or easier) than these, which I just came up with off the top of my head. But I think something along these lines is probably necessary to connect the ex vivo mechanism to its in vivo impact. A model would definitely not suffice.

We thank the reviewer for their thoughtful and constructive comments regarding the specificity of the in vivo experiments in Figure 7 and the connection between our ex vivo

and *in vivo* findings. We agree that the *in vivo* evidence for the involvement of the *ex vivo*-characterized mechanisms remained rather indirect. Indeed, more specific manipulations – ideally mGluRI or TASK/TREK conditional knockouts, would be the way to go about to strengthen our claims for these mechanisms. However, we feel – and hope that the reviewer will agree with us, that such a series of experiments warrants a project of its own and falls out of the scope of the current study. Nonetheless, we have taken up the reviewer's suggestion to provide a better link between the *ex vivo* and *in vivo* data by conducting additional *in vivo* whole-cell patch-clamp recordings on superficial pyramidal neurons during whisking stimulation (Extended Data Fig. 17 and cf. last point of Reviewer 1). In these new experiments, we have observed that superficial layer 2 pyramidal neurons (putatively BT cells, based on a qualitative morphological assessment) exhibit significant depolarizations upon activation of mGluRI, accompanied by an increase in membrane resistance. These results are consistent with our *ex vivo* findings and support the role of mGluR-mediated mechanisms in BT neurons during behaviorally relevant inputs.

15. The authors claim that *in vivo* calcium imaging confirms the presence of mGluR1-dependent modulation of inputs, but the MCPG/DHPG pharmacology reveals mixed effects. Of note, only 13% of upper L2/3 and <1% of lower L2/3 neurons decreased their activity upon application of antagonist MCPG. The authors should explain why the effect is so mixed, especially when their argument hinges on mGluR1 increasing dendritic excitability.

Indeed, the MCPG/DHPG pharmacology has mixed effects. This may be attributed to several factors. First, in the calcium imaging experiment we could not distinguish between the morphological subtypes and only base ourselves on superficial or deeper layers which will contain heterogeneous populations of L2/3 cells (also see Reviewer 2, point 18, below). Second, L2/3 in S1 are notorious for their sparsity in spiking under baseline conditions (<0.5 Hz; De Kock and Sackmann, 2009). This could be a major reason for why it was difficult to consistently observe decreases in calcium activity upon MCPG administration. This is also partly due to their highly variable sensitivities to whisking and brain-state-related network activity, and their variable connectivity rates. For example, even though L2 cells may receive more input from cortico-cortical circuits in L1 than the L3 cells, not all L2 cells will receive inputs from all these afferents to the same extent, and neither will the strength and timing of these inputs be the same. Together, this makes their activity rates capricious to start with, and the modulation thereof variable. Third, as the reviewer correctly pointed out under point 14, mGluRI are likely expressed at a variety of post and presynaptic sites in the cortical network and at those sites they are linked to a variety of intracellular mechanisms, which may produce various opposing effects to the modulation of excitation that is induced in the imaged neurons. Fourth, it is important to note that MCPG, as a metabotropic glutamate receptor antagonist, may not solely target mGluR1/5, but may also interact with mGluR2, which again may drive varying responses in the cortical network. Fifth, even within a specific neuronal population subtype, receptor expression, dendritic ion channel distribution, as well as intracellular signaling pathways may vary. Sixth, as also

discussed under point 14, the modulation of spiking activity might be an indirect readout of changes in a cell excitability, which depends on many factors. For example, a neuron that spikes upon strong driving inputs on its basal dendrites might not easily be modulated by mGluRI-related mechanisms on its apical dendrites. Altogether, these factors could have contributed to the variable responses upon mGluRI activation or inhibition. Importantly, the whole cell recordings do recapitulate the calcium imaging results in that the passive properties vary (in particular upon MCPG) but the overall trends are similar (in particular the increase in excitability upon DHPG). Long story short, we do agree with the reviewer that the effects are variable, we also think that this is not unexpected, and by no means invalidating of our hypothesis that mGluRI activation leads to increased excitability, preferentially in L2 neurons.

16. Related to point 2, Figure 7 is missing any measure of statistical significance of the deviation from baseline activity under MCPG and DHPG. Are the changes in activity statistically significant? While prediction power seems to be affected under the experimental conditions, the reduction of power under MCPG could also be explained by an increase in noise and not necessarily a decrease in activity. This could be easily addressed by showing how “noise” levels change before and after application of MCPG/DHPG, and whether such noise has any correlation with the PP (similar to Ext. Data Figure 11). Additionally, the authors do not explain why MCPG increases the overall activity in vivo when the opposite is expected based on their ex vivo pharmacology results.

We have revised Figure 7e and 7g to more effectively illustrate the deviation of population activity from baseline and to emphasize the differences between upper and lower L2/3 neurons following DHPG and MCPG injections. Additionally, we have included a statement indicating that both types of injections result in an overall increase in population-level activity.

To further strengthen our analysis, we introduced a new evaluation in which we calculated the noise power from the original activity traces before and after drug injections. Our findings indicate an increase in noise power in the calcium signal data, which may be attributed to local stress on the cortical tissue caused by the injection or the partial opacity of the drug solution interfering with fluorescence collection. To determine whether this noise increase influenced the decoding analysis, we compared the change in prediction power for each neuron with the change in noise power for both

DHPG and MCPG conditions. The analysis revealed no correlation between these variables, confirming that the decoding results were unaffected by noise levels in the data (see Extended Data Fig.15).

Extended Data Fig. 15 | The prediction power measurements are not sensitive to noise levels in the calcium activity traces.

a Comparison of the noise power (see Methods) in baseline to after cortical injection of DHPG. This indicates that the noise level during calcium imaging slightly increased after DHPG injection. **b** Same comparison after MCPG injection. **c** Comparison of the change in prediction power (PP) (after DHPG injection - baseline) to the change in noise power. No correlation was observed between the change in noise level of the measurement and the change in PP value. **d** Same analysis for the MCPG condition. The noise power was calculated as follows: the calcium activity signal was mean centered to remove any DC offset. It was then transformed into the frequency domain using the Fast Fourier Transform (FFT), and the Power Spectral Density (PSD) was computed.

17. The random forests decoder employed suggests increasing the gain of BT neurons through specific thalamocortical afferents increases the correlation between snout movements and activity. If the authors were to demonstrate a behavioral output difference (like an increase in discriminability between whisker stimulation frequencies or something of that sort), the impact of the novel BT-PoM depolarization mechanism would be stronger.

We thank the reviewer for this very good suggestion, which we believe would further strengthen the impact of our findings. However, we feel that investigating the behavioral consequences of the BT-POM depolarization mechanism would require a dedicated project due to the complexity of the experiments involved. This remains an exciting direction for future research.

18. Were any structural reconstructions from the *in vivo* calcium imaging experiments performed to confirm that imaging at 100 μm and 300 μm is an accurate proxy for imaging almost exclusively BT and ST cells? Could this explain some of the mixed effects in the *in vivo* pharmacology?

We thank the reviewer for raising this important point. While no structural reconstructions were performed on the *in vivo* calcium imaging experiments, our *in vivo* patch-clamp experiments, which were targeted to L2, included morphological identification of recorded neurons. The reconstructions reveal distinct morphological characteristics that align with the classifications of BT cells observed *in vitro* and the literature, i.e. a broad apical dendritic span, with often an asymmetrical orientation towards the pia (Reviewer Fig. 1). This supports the finding in Figure 1 that BT cells are biased towards L2. Hence, it is reasonable to expect that the imaging experiment captures mostly BT neurons. However, as is clearly visible in Fig. 1e and Extended Data Fig. 4, the depth of ST and BT cells is highly variable and overlapping, and this could have clearly contributed to the diluted effects observed in the *in vivo* pharmacology (also discussed in Reviewer 2, point 15). To outline this aspect, we originally stated in line 362 that: “For the analysis, we assumed that the population of BT neurons is enriched in upper L2/3 while the location of ST neurons is more biased towards deeper L2/3 (Fig. 1e).”

a

b

Reviewer Fig. 1. Morphology of superficial L2/3 neurons *in vivo*.

a Maximum intensity projection (MIP) from z-stacks acquired in 2-photon imaging of a L2/3 neuron patch-loaded with Alexa Fluor 488. Left images show the MIP in the lateral (x,y) and axial axes (x,z). The right image shows the same MIP x,y but color-coded by imaging depth. The neuron displays a typical broad-tufted (BT) morphology with an extensive and bi-tufted dendritic arborization. This example neuron is displayed in Extended Data Fig. 17b. **b** Three other examples of MIP from BT neurons. Scale bars: 50 μm .

19. Do the authors describe the epochs used for prediction of snout movements? Was the entire dataset used? Was the prediction power stable across the recording?

For the prediction of snout movements, we used the entire recording durations before and after drug injections (10 minutes of recordings). We have now included this important information in the main text (line 358). While it is likely that the prediction power fluctuates over time for individual neurons (as seen in Fig. 7j,l), our goal was to detect changes in prediction powers over long time periods across populations of neurons (Fig. 7k,m). The fact that the application of DHPG and MCPG bidirectionally affect the overall prediction power suggests that mGluR modulation shapes movement-related neural dynamics rather than just an effect of time which would unidirectionally affect the prediction power.

In addition, we performed a new analysis to check that the prediction power measurements were not sensitive to noise levels in the calcium activity traces (see Reviewer 2, point 16). While we quantified that the noise level changed after drug injections, no correlation was observed between the change in noise level and the change in prediction power (Extended Data Fig. 15), further suggesting that the observed effects could be attributed to the mGluR modulation.

20. Line 298 appears to incorrectly references Fig 6c.

Corrected

Reviewer #3 (Remarks to the Author):

In this very interesting manuscript Brandalise et al. have dissected the effects of thalamocortical projections stimulation on layer 2/3 pyramidal neurons apical dendrites from the mouse barrel cortex. They have used a panoply of state-of-the-art techniques, as optogenetics and dendritic patch-clamp recordings ex vivo (slices), plus 2-photon laser scanning microscopy of calcium activity in vivo, combined with pharmacology. Their findings indicate that higher-order thalamocortical inputs are able to modify L2/3 neurons excitability in a cell-type and input-selective mode. NMDAR spikes and plateau potentials (caused by mGluRI modulation of K2P channels activity) are the main mechanisms associated with this increased dendritic excitability. The in vivo experiments are in nice agreement with their ex vivo results, as upper L2/3 neurons (mainly BT) bidirectional activity changes (produced by pharmacological manipulation of mGluRI signaling) correlate with the mice snout movements.

Overall, the conclusions of this paper are well supported by the data, the manuscript is well-written, nicely organized, the quality and quantity of work is considerable, and these findings should be of broad interest for the scientific community.

I only have some minor points that should be addressed or clarified.

- K2P channels are involved in the regulation of resting membrane potential, as the authors mention. When the experimenters blocked these channels (figure 5), did they observed a depolarization of the membrane potential in their recordings, comparing with the control condition? To some extent, this blockade could account maybe as a kind of “sustained” plateau potential, and maybe an increase in spontaneous activity has been observed in those recordings.

We thank the reviewer for raising this interesting question. To address this point, we have now included Extended Data Fig. 11, which provides examples of recorded dendrites where K2P channels were pharmacologically blocked. In response to this manipulation, we observed a transient, reversible depolarization accompanied by an increase in input resistance. This additional dataset supports our conclusions and provides further insight into the role of K2P channels in regulating dendritic excitability.

Extended Data Fig. 11 | Effect K2P blockers on the resting membrane potential (related to Fig. 5).

a Representative dendritic recording showing the effect of a cocktail of K2P channel blockers, including bupivacaine (1 mM), QX-314 (1 mM), and barium (1 mM), as indicated by the red bar. Application of the blockers induced a depolarization of the membrane potential, reflecting a reduction in K2P-mediated resting potassium conductance and an increase in input resistance. The depolarization occurred without external stimulation, highlighting the contribution of K2P channels to maintaining resting membrane potential. Washout of the drug cocktail led to a restoration of the baseline membrane potential, indicating reversibility of the effect. **b** Example trace of a dendritic recording illustrating a DSDP during K2P blockade. Arrows indicate excitatory postsynaptic potentials (EPSPs) occurring on top of the plateau potential. The increase in EPSP activity suggests enhanced synaptic responsiveness during the plateau state compared to baseline conditions. **c** Quantification of EPSP frequency under baseline conditions and during plateau potentials. The frequency of EPSPs was significantly higher during plateau potentials compared to baseline (in baseline: 2.3 ± 0.32 Hz; on top of plateaus: 3.7 ± 0.35 Hz, means \pm s.e.m.; $n = 8$ neurons, $P = 0.004$, paired t-test). This finding highlights a potential role for K2P channel activity in modulating synaptic input integration during altered membrane potential states.

- The authors use a specific mGlu1 receptor antagonist (LY367385) and a generic group I / group II mGluRs antagonist (MCPG). Did they try a specific mGlu5 receptor antagonist as MPEP or MTEP?

We thank the reviewer for this valuable suggestion. Following the reviewer's recommendation, we have performed additional experiments using the selective mGluR5 receptor antagonist MPEP. Interestingly, we observed no significant effect of MPEP application on the occurrence of the events (see updated Fig. 6 below). Yet, they did disappear upon subsequent addition of LY367385, the selective mGluR1 antagonist. These findings suggest that the observed effects are primarily mediated by mGluR1 receptors, which interestingly align with the expression profiles of these receptors in our patch-seq experiments and the Allen Mouse Brain mRNA expression atlas (see reviewer 2, point 1). The MPEP results are now included in Fig. 6d,e,

Fig. 6 | POm-mediated DSDPs are mediated by the activation of mGluR1s.

a Dendritic recordings of a BT neuron during the stimulation of POm before and after the bath application of LY367385 (50 μ M), a selective mGluR1 blocker. LY367385 prevented the generation of DSDPs. **b** The DSDP frequency was mostly prevented in the presence of LY367385 (50 μ M; $n = 4$, from 4 mice, $P = 0.003$, paired t-test) as well as in the presence of the generic mGluR1 blocker MCPG (500 μ M; $n = 3$, from 3 mice, $P = 0.03$, paired t-test). **c** Together, the mGluR1 blockers LY367385 and MCPG significantly decrease the membrane resistance of the recorded dendrites ($n = 7$, from 4 mice, $P = 0.04$, paired t-test). **d** Dendritic recordings of BT neurons in the presence of MPEP, a selective mGluR5 blocker, did not significantly change the DSDP frequency following POm afferent photostimulations (10 μ M; $n = 5$, from 5 mice, $P = 0.25$, paired t-test). **e** In another set of recordings, LY367385 was subsequently added to the MPEP containing ACSF solution which significantly reduced the DSPD frequency ($n = 6$, from 4 mice, $P = 0.029$, paired t-test). DSDP frequency did not significantly recover after LY367385 washout ($P = 0.14$;

MPEP pre vs MPEP post, $P = 0.08$; paired t-test). **f** Dendritic recording of a BT neuron during the stimulation of POM afferents in the presence of GDP- β -S (1 mM), a G-protein activity blocker, in the intracellular solution. Within the first minute after break-in, DSDPs could be observed but then disappeared after 3 minutes, consistent with the dialysis of the drug. **g** The frequency of DSDPs was largely reduced after intracellular dialysis of GDP- β -S ($n = 4$, from 4 mice, $P = 0.003$, paired t-test). **h** The dialysis of GDP- β -S did not significantly reduce the membrane resistance of the dendrite ($n = 4$, from 4 mice, $P = 0.066$, paired t-test).

- Panels c and e from Extended Data Fig. 9 show input resistance changes and the average and SEM data are included in the figure legend. The numeric data could be included as well for input resistance plots in Fig. 5f and Fig. 6c,f. It seems clear that blocking K2P channels (Fig. 5f) will increase the dendritic input resistance at all times. However, it is not as clear for me if the blocking of mGluR1s (Fig. 6c) should be reflected in an input resistance decrease. Blocking the mGluR1s will prevent the K2P closing in those conditions that promote plateau potentials (i.e., POM stimulation). But during the input resistance protocol (exemplified in Extended Data Fig.1, panel d), that stimulation is not happening, as I understand. Moreover, in Fig. 6f, dialysis of GDP-β-S did not significantly reduce the input resistance of the dendrite. P values for 6c and 6f are close to 0.05 in both cases, but the change of this parameter, if any, it would be probably very small.

The reviewer correctly points out that the readout of reduced input resistance in the mGluR1s blocking experiment is tricky. Indeed, in these experiments cells will likely operate under very low baseline levels of mGluR1 activation, particularly because in acute slice preparations the spontaneous activity levels are relatively low compared to *in vivo* conditions. As a result, blocking mGluR1s may exert minimal effects on the passive properties of the recorded cell, specifically the input resistance – which is what we see indeed (Fig. 6c). Nonetheless, probably due to interference with some background activity of this pathway, the effects become noticeable.

Additionally, as the reviewer has pointed out, the dialysis of GDP-β-S in Fig. 6f did not significantly reduce the input resistance of the dendrite (<1 min : $211 \pm 46 \text{ M}\Omega$; >3 min : $194 \pm 51 \text{ M}\Omega$; means \pm s.e.m.; $n = 4$, $P = 0.066$, paired t-test). This is likely because GDP-β-S blocks all G-proteins non-selectively, which may introduce aspecific effects. While POM stimulation or DHPG application preferentially acts on G-proteins coupled to mGluR1s, the global blockade of G-protein signaling with GDP-β-S may not entirely reflect the specific pathway modulated by mGluR1s.

To further clarify and enhance transparency, we have now included the numeric input resistance data for Fig. 5f and Fig. 6c,f (now Fig. 6c,h), as suggested, and these values are presented in the respective figure legends. We hope these additional details and considerations adequately address the reviewer's concerns.

- Extended Data Fig. 10 could be improved performing for the data in panel b the same quantification done in Fig. 4c. The scatter plot for the POM stim. initial condition (previous to TBOA application) could be added as well.

We thank the reviewer for this valuable suggestion. As requested, we have now added the quantification for the data in panel b of Extended Data Fig. 10 (now Extended Data Fig. 14), consistent with the analysis performed in Fig. 4c. Additionally, the scatter plot for the POM stimulation initial condition (prior to TBOA application) has been included to enhance clarity and completeness. These updates are reflected in the revised figure.

- There is some degree of dissimilarity between some sentences in Extended Data Fig. 11 legend and the main text. In the figure legend, it is said that for the DHPG dataset, “little to no correlation were observed for the upper and lower L2/3 neurons”; in the main text, “the PP of upper L2/3 neurons under control conditions moderately correlated with the magnitude of increase in activity under DHPG”. The R value for upper L2/3 is 0.1 (DHPG), and R=-0.11 for lower L2/3 in MCPG. However, for lower L2/3 (MCPG), both main text and figure legend claim little to no correlation. Maybe the “moderately correlated” rating for upper L2/3 (DHPG) in the main text could be tempered.

We have modified the sentence according to the reviewer’s suggestion.

Minor details (typos, etc.):

- Lines 108-109: “(Extended Data Fig. 1b,c)” → actually corresponds to “(Fig. 1c,d)” Fixed
- Line 196: “distribution of the evoked PSP amplitudes” → the parameter mentioned in figure 3 and Extended Data Fig. 6 as “event size” or “amplitude” corresponds to PSP peak area (mV·ms) rather than PSP peak amplitude (mV), as I understand from the figures and values. Maybe this could be clarified in the text. Fixed
- Line 264: “at -120 mV” → “at -125 mV” (measurements were done at -60, -100 and -125 mV) Fixed
- Line 298: “(Fig. 6c)” → actually corresponds to “(Fig. 6d)” Fixed
- Line 324: “(Extended Data Fig. 10A,B)” → “(Extended Data Fig. 10a,b)”, lowercase for the panels letters Fixed
- Line 901: “500 μm, Tocris Bioscience” → “500 μM, Tocris Bioscience” Fixed
- Line 1175: “The bath application of 50 mM APV” → “The bath application of 50 μM APV” Fixed
- Line 1212: “S1intracortical vs. VPM: P = 0.055” → This comparison/cumulative distribution plot is not shown in panel g, but panel j later, in line 1217. Fixed
- Lines 1312-1314: “The passive properties were assessed by analyzing the changes in current to a hyperpolarizing step from -70 mV to -80 mV (bottom)” → it should say “(right)” because it is placed at the right side of panel d. Fixed
- Page 61, Extended Data Fig. 9: in panel a, “10 μm” should be changed to “10 μM” (appears twice). Fixed